# OFF-POLICY MAXIMUM ENTROPY RL WITH FUTURE STATE AND ACTION VISITATION MEASURES

## ABSTRACT

We introduce a new maximum entropy reinforcement learning framework based on the distribution of states and actions visited by a policy. More precisely, an intrinsic reward function is added to the reward function of the Markov decision process that shall be controlled. For each state and action, this intrinsic reward is the relative entropy of the discounted distribution of states and actions (or features from these states and actions) visited during the next time steps. We first prove that an optimal exploration policy, which maximizes the expected discounted sum of intrinsic rewards, is also a policy that maximizes a lower bound on the state-action value function of the decision process under some assumptions. We also prove that the visitation distribution used in the intrinsic reward definition is the fixed point of a contraction operator. Following, we describe how to adapt existing algorithms to learn this fixed point and compute the intrinsic rewards to enhance exploration. A new practical off-policy maximum entropy reinforcement learning algorithm is finally introduced. Empirically, exploration policies have good state-action space coverage, and high-performing control policies are computed efficiently.

## 1 INTRODUCTION

Many challenging tasks where an agent makes sequential decisions have been solved with reinforcement learning (RL). Examples range from playing games (Mnih et al., 2015; Silver et al., 2017), or controlling robots (Kalashnikov et al., 2018; Haarnoja et al., 2018a), to managing the energy systems and markets (Boukas et al., 2021; Aittahar et al., 2024). In practice, many RL algorithms are applied in combination with an exploration strategy to achieve high-performance control. Assuming the agent takes actions in a Markov decision process (MDP), these exploration strategies usually consist in providing intrinsic reward bonuses to the agent for achieving certain behaviors. Typically, the bonus enforces taking actions that reduce the uncertainty about the environment (Pathak et al., 2017; Burda et al., 2018; Zhang et al., 2021b), or actions that enhances the variety of states and actions in trajectories (Bellemare et al., 2016; Lee et al., 2019; Guo et al., 2021; Williams & Peng, 1991; Haarnoja et al., 2019). In many of the latter methods, the intrinsic reward function is the entropy of some distribution over the state-action space. Optimizing jointly the reward function of the MDP and the intrinsic reward function, in order to eventually obtain a high-performing policy, is called Maximum Entropy RL (MaxEntRL) and was shown effective in many problems.

The reward of the MDP was already extended with the entropy of the policy in early algorithms (Williams & Peng, 1991) and was only later called MaxEntRL (Ziebart et al., 2008; Toussaint, 2009). This particular reward regularization provides substantial improvements in robustness of the resulting policy (Ziebart, 2010; Husain et al., 2021; Brekelmans et al., 2022) and provides a learning objective function with good smoothness and concavity properties (Ahmed et al., 2019; Bolland et al., 2023). Several commonly used algorithms can be named, like soft Q-learning (Haarnoja et al., 2017; Schulman et al., 2017) and soft actor-critic (Haarnoja et al., 2018b; 2019). This MaxEntRL framework nevertheless only rewards the randomness of actions and neglects the influences of the policy on the visited states, which, in practice, leads to inefficient exploration in many scenarios.

In order to enhance exploration, Hazan et al. (2019) were first to propose to intrinsically motivate agents to have a uniform discounted visitation measure over states. Several works have afterwards been developed to maximize the entropy of the discounted state visitation measure and the stationary state visitation measure. For discrete state and action spaces, optimal exploration policies, which

maximize the entropy of these visitation measures, can be computed to near optimality with off-policy tabular model-based RL algorithms (Hazan et al., 2019; Mutti & Restelli, 2020; Tiapkin et al., 2023). For continuous state and action spaces, alternative methods rely on $k$ nearest neighbors to estimate the density of the visitation measure of states (or features built from the states) and compute the intrinsic rewards, which can afterwards be optimized with any RL algorithm (Liu & Abbeel, 2021; Yarats et al., 2021; Seo et al., 2021; Mutti et al., 2021). These methods require sampling new trajectories at each iteration, they are on-policy, and estimating the intrinsic reward function is computationally expensive. Some other methods rely on parametric density estimators to reduce the computational complexity and share information across learning steps (Lee et al., 2019; Guo et al., 2021; Islam et al., 2019; Zhang et al., 2021a). The additional function approximator is typically learned on-policy by maximum likelihood estimation based on batches of truncated trajectories. Worth noticing methods have adapted this MaxEntRL framework to maximize entropy of states visited in single trajectories instead of on expectation over trajectories (Mutti et al., 2022; Jain et al., 2024). When large and/or continuous state and action spaces are involved, relying on parametric function approximators is likely the best choice. Nevertheless, existing algorithms are on-policy. They require sampling new trajectories from the environment at (nearly) every update of the policy, and can not be applied using a buffer of arbitrary transitions, in batch-mode RL, or in continuing tasks. Furthermore, learning the discounted visitation measure is more desirable than learning the stationary one, but may be challenging in practice due to the exponentially decreasing influence of the time step at which states are visited (Islam et al., 2019).

The main contribution of this paper is to introduce a MaxEntRL framework relying on a new intrinsic reward function, for exploring effectively the state and action spaces, that also alleviates the previous limitations. In this new MaxEntRL framework, for each state and action, the intrinsic reward function is the relative entropy of the discounted distribution of states and actions (or features from these states and actions) visited during the next time steps. We first prove that a policy maximizing the expected discounted sum of these rewards is also one that maximizes a lower bound on the state-action value function of the MDP under some assumptions. In addition, we prove that the visitation distribution used in the intrinsic reward function definition is the fixed point of a contraction operator. Existing RL algorithms can integrate an additional learning step to approximate this fixed point off-policy, using N-step state-action transitions and bootstrapping the operator. It is then possible to approximate the intrinsic reward function and learn a policy maximizing the extended rewards with the adapted algorithm. We illustrate this methodology on off-policy actor-critic (Degris et al., 2012). The resulting MaxEntRL algorithm is off-policy, computes efficiently exploration policies with uniform discounted state visitation and high-performing control policies.

The visitation measure of future states and actions, which we use to extend the reward function in this article, has a well-established history in the development of RL algorithms. It was popularized by Janner et al. (2020), who learned the distribution of future states as a generalization of the successor features (Barreto et al., 2017). He demonstrated that this distribution allows to express the state-action value function by separating the influence of the dynamics and the reward function, and that it could be learned off-policy exploiting its recursive expression. Several algorithms have been proposed to learn this distribution, either by maximum likelihood estimation (Janner et al., 2020), by contrastive learning (Mazoure et al., 2023b), or using diffusion models (Mazoure et al., 2023c). These distributions of future states and actions have found applications in goal-based RL (Eysenbach et al., 2020; 2022), in offline pre-training with expert examples (Mazoure et al., 2023a), in model-based RL (Ma et al., 2023), or in planning (Eysenbach et al., 2023). We are the first to integrate them into a MaxEntRL framework for enhancing exploration through the policy learning.

The manuscript is organized as follows. In Section 2, the problem of computing optimal policies is reminded and a general MaxEntRL framework is formulated. In Section 3, we introduce MaxEntRL with conditional state-action visitation probability and show how policies can be computed in this framework. Finally, in Section 4 we present experimental results and conclude in Section 5.

## 2 BACKGROUND AND PRELIMINARIES

### 2.1 MARKOV DECISION PROCESSES

This paper focuses on problems in which an agent makes sequential decisions in a stochastic environment (Sutton & Barto, 2018). The environment is modeled with an infinite-time Markov decision process (MDP) composed of a state space $\mathcal{S}$, an action space $\mathcal{A}$, an initial state distribution $p_0$, a transition distribution $p$, a bounded reward function $R$, and a discount factor $\gamma \in [0, 1)$. Agents interact in this MDP by providing actions sampled from a policy $\pi$. During this interaction, an initial state $s_0 \sim p_0(\cdot)$ is first sampled, then, the agent provides at each time step $t$ an action $a_t \sim \pi(\cdot|s_t)$ leading to a new state $s_{t+1} \sim p(\cdot|s_t, a_t)$. In addition, after each action $a_t$ is executed, a reward $r_t = R(s_t, a_t) \in \mathbb{R}$ is observed. We denote the expected return of the policy $\pi$ by

$$J(\pi) = \mathbb{E}_{\substack{s_0 \sim p_0(\cdot) \\ a_t \sim \pi(\cdot|s_t) \\ s_{t+1} \sim p(\cdot|s_t, a_t)}} \left[ \sum_{t=0}^{\infty} \gamma^t R(s_t, a_t) \right] . \tag{1}$$

An optimal policy $\pi^*$ is a policy with maximum expected return.

### 2.2 MAXIMUM ENTROPY REINFORCEMENT LEARNING

In maximum entropy reinforcement learning (MaxEntRL) an optimal policy $\pi^*$ is approximated by maximizing a surrogate objective function $L(\pi)$, where the reward function from the MDP is extended by an intrinsic reward function. The latter is the (relative) entropy of a conditional distribution. A general definition of the MaxEntRL objective function is

$$L(\pi) = \mathbb{E}_{\substack{s_0 \sim p_0(\cdot) \\ a_t \sim \pi(\cdot|s_t) \\ s_{t+1} \sim p(\cdot|s_t, a_t)}} \left[ \sum_{t=0}^{\infty} \gamma^t \left( R(s_t, a_t) + \lambda R^{int}(s_t, a_t) \right) \right] , \tag{2}$$

where $R^{int}$ is the intrinsic reward function. As discussed in Section 1, different MaxEntRL frameworks exist, each defining as intrinsic reward the entropy of some particular distribution. We propose a generic formulation for the intrinsic reward, which to the best of our knowledge encompasses all existing frameworks from the litterature. Given a feature space $\mathcal{Z}$, a conditional distribution $q^\pi : \mathcal{S} \times \mathcal{A} \to \Delta(\mathcal{Z})$, depending on the policy $\pi$, and a relative measure $q^* \in \Delta(\mathcal{Z})$, the intrinsic reward function is

$$R^{int}(s, a) = -KL_z \left[ q^\pi(z|s, a) \| q^*(z) \right] = \mathbb{E}_{z \sim q^\pi(\cdot|s, a)} \left[ \log q^*(z) - \log q^\pi(z|s, a) \right] . \tag{3}$$

Importantly, the intrinsic reward function is (implicitly) dependent on the policy $\pi$ through the distribution $q^\pi$. We define an optimal exploration policy as a policy that maximizes the expected sum of discounted intrinsic rewards only. In practice, as soon as it is possible to generate samples from the distribution $q^\pi(z|s, a)$ and estimate their probabilities, the intrinsic reward equation (3) can be estimated by Monte-Carlo, and used in any existing RL algorithm to extend the MDP reward function. Note that a policy maximizing $L(\pi)$ is generally not optimal, due to the potential gap between the optimum of the return $J(\pi)$ and the optimum of the learning objective $L(\pi)$. This subject is inherent to exploration with intrinsic rewards and is extensively discussed by Bolland et al. (2024).

Many of the existing MaxEntRL algorithms rely on the entropy of the policy for exploring the action space (Haarnoja et al., 2018b; Toussaint, 2009). The feature space is then the actions space $\mathcal{Z} = \mathcal{A}$, and the conditional distribution is the policy $q^\pi(z|s, a) = \pi(z|s)$, for all $a$. Other algorithms focus on exploring the state space (Lee et al., 2019; Islam et al., 2019; Guo et al., 2021). The feature space is the state space $\mathcal{Z} = \mathcal{S}$. The conditional distribution $q^\pi(z|s, a)$ is either the marginal probability of states in trajectories of $T$ time steps, or the discounted state visitation measure, for all $s$ and $a$. In the literature, the relative measure $q^*(z)$ is usually a uniform distribution, and the relative entropy is computed as the differential entropy, i.e., by neglecting $\log q^*(z)$ in equation (3). In continuous spaces, the latter is ill-defined and other relative measures are sometimes used.

## 3 OFF-POLICY MAXENTRL WITH VISITATION DISTRIBUTIONS

### 3.1 DEFINITION OF MAXENTRL WITH CONDITIONAL VISITATION DISTRIBUTIONS

In the following, we introduce a new MaxEntRL framework based on the conditional state-action visitation probability measure $d^{\pi,\gamma}(s_\infty, a_\infty | s, a)$ and the conditional state visitation probability measure $d^{\pi,\gamma}(s_\infty | s, a)$

$$d^{\pi,\gamma}(s_\infty, a_\infty | s, a) = (1 - \gamma)\pi(a_\infty | s_\infty) \sum_{\Delta=1}^{\infty} \gamma^\Delta p_\Delta^\pi(s_\infty | s, a) \tag{4}$$

$$d^{\pi,\gamma}(s_\infty | s, a) = (1 - \gamma) \sum_{\Delta=1}^{\infty} \gamma^\Delta p_\Delta^\pi(s_\infty | s, a) , \tag{5}$$

where $p_\Delta^\pi$ is the transition probability in $\Delta$ time steps with the policy $\pi$. The distribution from equation (4) can be factorized as a function of the distribution from equation (5) such that $d^{\pi,\gamma}(s_\infty, a_\infty | s, a) = \pi(a_\infty | s_\infty) d^{\pi,\gamma}(s_\infty | s, a)$. The conditional state (respectively, state-action) visitation probability distribution measures the states (respectively, states and actions) that are visited on expectation over infinite trajectories starting from a state and an action. Both distributions generalize the state visitation probability measure (Manne, 1960).

In our MaxEntRL framework, the feature space $\mathcal{Z}$ and a feature distribution $h : \mathcal{S} \times \mathcal{A} \to \Delta(\mathcal{Z})$ are assumed provided. The intrinsic reward is computed according to equation (3), for any relative measure $q^*$, with conditional distribution

$$q^\pi(z | s, a) = \int h(z | s_\infty, a_\infty) d^{\pi,\gamma}(s_\infty, a_\infty | s, a) \, ds_\infty \, da_\infty . \tag{6}$$

Optimal exploration policy are here intrinsically motivated to take actions so that the discounted visitation measure of future features is distributed according to $q^*$ in each state and for each action. It allows to select features that must be visited during trajectories according to prior knowledge about the problem if any. Furthermore, samples can easily be generated from the distribution equation (6) by sampling a state $s_\infty \sim d^{\pi,\gamma}(\cdot | s, a)$, an action $a_\infty \sim \pi(\cdot | s_\infty)$, and finally sampling a feature $z \sim h(\cdot | s_\infty, a_\infty)$. Similarly, the probability of this sample can be estimated solving the integral numerically using samples $s_\infty \sim d^{\pi,\gamma}(\cdot | s, a)$ and $a_\infty \sim \pi(\cdot | s_\infty)$.

Let us finally relate MaxEntRL with this new intrinsic reward function to the maximization of a lower bound on the state-action value function of the MDP (computed without intrinsic reward bonuses). We rely on the bound on the relative suboptimality gap of the state-action value function of a policy $\pi$, i.e., $(Q^{\pi^*}(s, a) - Q^\pi(s, a))/Q^{\pi^*}(s, a)$, using Theorem 1, proved in Appendix A.

**Theorem 1.** Let the reward function $R(s, a)$ be non-negative, and let $\pi$ be a policy with state-action value function $Q^\pi(s, a)$, then,

$$\frac{Q^{\pi^*}(s, a) - Q^\pi(s, a)}{Q^{\pi^*}(s, a)} \leq 1 - \exp\left(-\|\log d^{\pi,\gamma}(\cdot, \cdot | s, a) - \log d^{\pi^*,\gamma}(\cdot, \cdot | s, a)\|_\infty\right), \tag{7}$$

where $\|f\|_\infty = \sup_x |f(x)|$ is the $L_\infty$-norm of the function $f$.

Let us consider the MaxEntRL framework with the intrinsic reward function equation (6) where $\mathcal{Z} = \mathcal{S} \times \mathcal{A}$ and $h(z | s, a)$ is a dirac distribution centered in $z = (s, a)$, and with the relative measure $q^*(s, a)$. Let us apply the triangle inequality to the bound in Theorem 1. For any $\pi$, we get the bound

$$\frac{Q^{\pi^*}(s, a) - Q^\pi(s, a)}{Q^{\pi^*}(s, a)} \leq 1 - \exp\left(-\|\log d^{\pi,\gamma}(\cdot, \cdot | s, a) - \log q^*(\cdot, \cdot)\|_\infty\right.$$
$$\left. - \|\log d^{\pi^*,\gamma}(\cdot, \cdot | s, a) - \log q^*(\cdot, \cdot)\|_\infty\right). \tag{8}$$

According to equation (8), the relative suboptimality gap of the state-action value function of any policy $\pi$ depends on two error terms $\|\log d^{\pi,\gamma}(\cdot, \cdot | s, a) - \log q^*(\cdot, \cdot)\|_\infty$ and $\|\log d^{\pi^*,\gamma}(\cdot, \cdot | s, a) - \log q^*(\cdot, \cdot)\|_\infty$. The first can be minimized as a function of $\pi$ while the second is independent of the policy, and can thus not be reduced. Let us assume that an optimal exploration policy has zero

expected discounted sum of intrinsic rewards, and that the target measure and the visitation measures are smooth. Then, an optimal exploration policy also minimizes the bound on the gap in equation (8). Computing optimal exploration policies in the MaxEntRL framework we introduce can then be interpreted as a practical algorithm for computing a policy minimizing the upper bound in equation (8) provided a priori with the target. The quality of the optimal exploration policy is then only dependent on this a priori choice of target.

## 3.2 Learning Objective for Conditional Visitation Models

As explained in Section 2.2, the intrinsic reward can always be computed by sampling from the conditional distribution $q^\pi(z|s, a)$ and evaluating the probability of these samples. Furthermore, in the new MaxEntRL framework introduced in Section 3.1, the conditional distribution $q^\pi(z|s, a)$ can be computed based on samples of the conditional state visitation distribution $d^{\pi,\gamma}(s_\infty|s, a)$. In this section, we explain how the latter distribution can be approximated.

In order to estimate the conditional state visitation distribution, we first recall that this distribution is a fixed point of the operator $\mathcal{T}^\pi$ defined by Janner et al. (2020)

$$\mathcal{T}^\pi d^{\pi,\gamma}(s_\infty|s, a) = (1 - \gamma)p(s_\infty|s, a) + \gamma \mathop{\mathbb{E}}_{\substack{s' \sim p(\cdot|s,a) \\ a' \sim \pi(\cdot|s')}} [d^{\pi,\gamma}(s_\infty|s', a')] \ . \tag{9}$$

Theorem 2, proved in Appendix A, states that the operator $\mathcal{T}^\pi$ is a contraction mapping, which furthermore implies the uniqueness of its fixed point. Assuming the result of the operator could be computed (or estimated), the fixed point could theoretically also be computed by successive application of this operator. The computation of that fixed point would then allow computing the conditional state-action visitation distribution, and the intrinsic reward function.

**Theorem 2.** The operator $\mathcal{T}^\pi$ is $\gamma$-contractive in $\bar{L}_n$-norm, where $\bar{L}_n(f)^n = \sup_y \int f(x|y)^n \, dx$.

In practice, computing the result of the operator $\mathcal{T}^\pi$ (and $(\mathcal{T}^\pi)^N$ after $N$ applications) may be intractable when large state and action spaces are at hand or when these spaces are continuous. It furthermore requires having a model of the MDP. A common approach is then to rely on a function approximator $d_\psi$ to approximate the fixed point. Furthermore, similarly to TD-learning methods (Sutton & Barto, 2018), Theorem 2 suggests to optimize the parameters of this model $d_\psi$ to minimize the residual of the operator, measured with an $\bar{L}_n$-norm for which the operator is $\gamma$-contractive. With this metric, estimating the residual requires estimating the transition function (Janner et al., 2020), and cannot be trivially minimized by stochastic gradient descent using transitions from the environment. We therefore propose to solve as surrogate a minimum cross-entropy problem, in which stochastic gradient descent can be applied afterwards. For any policy $\pi$, the distribution is approximated with a function approximator $d_\psi$ with parameter $\psi$ optimized to solve

$$\arg\min_\psi \mathop{\mathbb{E}}_{\substack{s,a \sim g(\cdot,\cdot) \\ s_\infty \sim (\mathcal{T}^\pi)^N d_\psi(\cdot|s,a)}} [-\log d_\psi(s_\infty|s, a)] \ , \tag{10}$$

where $g$ is an arbitrary distribution over the state and action space, and where $N$ is any positive integer. This optimization problem may be related to minimizing the KL-divergence instead of an $\bar{L}_n$-norm (Bishop & Nasrabadi, 2006).

Let us make explicit how samples from the distribution $(\mathcal{T}^\pi)^N d_\psi(s_\infty|s, a)$ are generated using the MDP. By definition of the operator $\mathcal{T}^\pi$, the distribution $(\mathcal{T}^\pi)^N d_\psi(s_\infty|s, a)$ is a mixture where the probability of samples is a weighted sum composed of the $N$ first multi-step transition probabilities in the MDP and the conditional state visitation model

$$(\mathcal{T}^\pi)^N d_\psi(s_\infty|s, a) = \left( \sum_{\Delta=1}^N (1 - \gamma)\gamma^{\Delta-1} p_\Delta^\pi(s_\infty|s, a) \right) + \gamma^N \mathop{\mathbb{E}}_{\substack{s' \sim p_N^\pi(\cdot|s,a) \\ a' \sim \pi(\cdot|s')}} [d_\psi(s_\infty|s', a')] \quad (11)$$

$$= \sum_{\Delta=1}^\infty Geom(\Delta|1 - \gamma) b_{\psi,\pi}^\beta(s_\infty|s, a, \Delta)\big|_{\beta=\pi} \ , \tag{12}$$

where $Geom(\Delta|1-\gamma)$ is the probability of the result $\Delta$ from a geometric distribution of parameter $1-\gamma$, and where

$$b^{\beta}_{\psi,\pi}(s_{\infty}|s,a,\Delta) = \begin{cases} p^{\beta}_{\Delta}(s_{\infty}|s,a) & \text{if } \Delta \leq N \\ \mathbb{E}_{\substack{s' \sim p^{\beta}_N(\cdot|s,a) \\ a' \sim \pi(\cdot|s')}} [d_{\psi}(s_{\infty}|s',a')] & \text{if } \Delta > N \end{cases}. \tag{13}$$

Sampling from $(\mathcal{T}^{\pi})^N d_{\psi}(s_{\infty}|s,a)$ consists in sampling from the mixture. First, $\Delta$ is drawn from a geometric distribution of parameter $1-\gamma$. Second, a state is sampled as $s_{\infty} \sim p^{\pi}_{\Delta}(\cdot|s,a)$ if $\Delta \leq N$ or as $s_{\infty} \sim d_{\psi}(\cdot|s',a')$ otherwise; where $s' \sim p^{\beta}_N(\cdot|s,a)$ and $a' \sim \pi(\cdot|s')$.

Finally, we reformulate the problem equation (10) such that it can be estimated from trajectories sampled from any policy $\beta$ in the MDP. To that end, we apply importance weighting and get the equivalent optimization problem

$$\arg\min_{\psi} \mathbb{E}_{\substack{s,a \sim g(\cdot,\cdot) \\ \Delta \sim Geom(\cdot|1-\gamma) \\ s_{\infty} \sim b^{\beta}_{\psi,\pi}(\cdot|s,a,\Delta)}} \left[ -\frac{b^{\pi}_{\psi,\pi}(s_{\infty}|s,a,\Delta)}{b^{\beta}_{\psi,\pi}(s_{\infty}|s,a,\Delta)} \log d_{\psi}(s_{\infty}|s,a) \right]. \tag{14}$$

In the particular cases where $\beta = \pi$ or where $N = 1$, the importance weight simplifies to one, otherwise it can be simplified to a (finite) product of ratios of policies. We do not delve into more details as we neglected this factor in Section 3.3 when using stochastic gradient descent.

## 3.3 PRACTICAL MAXENTRL EXPLORATION ALGORITHMS

Existing algorithms can finally be adapted to implement the MaxEntRL framework from Section 3.1 without substantial modifications. An additional learning step is integrated to update the conditional state visitation model $d_{\psi}$ to minimize the objective from equation (14). The intrinsic reward can then be computed and this intrinsic reward and the MDP reward are jointly optimized.

**Learning the conditional state visitation.** At each learning iteration of the RL algorithm, the parameter $\psi$ of the visitation model $d_{\psi}$ is also updated. First, the objective function from equation (14) is estimated by Monte-Carlo using trajectories simulated in the MDP from an arbitrary policy $\beta$. The importance weight is neglected. Second, this estimate is differentiated and the parameter $\psi$ is updated by gradient descent steps. Formally, let us assume that $N$-step transitions $(s_{t:t+N}, a_{t:t+N-1})$ computed from an arbitrary policy $\beta$ are sampled and stored as a batch or in a replay buffer $\mathcal{D}$. The state action pair $(s_t, a_t)$ is distributed according to the distribution $g$, which depends on the generation procedure of the transitions. The visitation distribution $d_{\psi}$ corresponding to the optimized policy $\pi_{\theta}$ is iteratively updated performing stochastic gradient descent steps on the loss function

$$\mathcal{L}(\psi) = -\sum_{s_t,a_t \in \mathcal{D}} \log d_{\psi}(s_{\infty}|s_t,a_t), \tag{15}$$

where $\Delta$ is sampled from a geometric distribution of parameter $1-\gamma$. The state $s_{\infty} = s_{t+\Delta}$ is available in the batch or replay buffer if $\Delta \leq N$, or $s_{\infty} \sim d_{\psi'}(\cdot|s_{t+N}, a_{t+N'})$ is bootstrapped otherwise; where $a_{t+N'} \sim \pi(\cdot|s_{t+N})$ and where $\psi'$ is the target network parameter. In the latter bootstrapping operation, an action is sampled from the policy $\pi$, making the algorithm off-policy.

In practice, the gradients generated by differentiating this loss function are biased estimates of the gradients from the objective function equation (14). The influence of the parameter $\psi$ on the probability of the sample $s_{\infty}$ is neglected, i.e., the partial derivative of $(\mathcal{T}^{\pi})^N d_{\psi}(s_{\infty}|s_t,a_t)$ with respect to $\psi$ is neglected, and a target network is used. This is analogue to SARSA and TD-learning strategies (Sutton & Barto, 2018). Furthermore, the importance weights from equation (14) is neglected too. It introduces a dependency of the distribution $d_{\psi}$ on the policy $\beta$ for the $N$ first steps, which is again similar to multi-step SARSA and multi-step TD-learning approaches.

**Computing the intrinsic reward.** The final modification to adapt existing RL algorithms to this new MaxEntRL framework is to compute the intrinsic reward function every time the reward is processed by the algorithm. For that step, the entropy of the distribution $q^{\pi}$ is estimated with

$$R^{int}(s_t,a_t) = \log q^*(z_t) - \log q^{\pi}(z_t|s_t,a_t) \tag{16}$$

where $z_t \sim q^\pi(\cdot|s_t, a_t)$ and where $q^\pi(z_t|s_t, a_t)$ is approximated by Monte-Carlo integration of the integral equation (6). Note that this integral may have a closed-form depending on the choice of feature space $\mathcal{Z}$ and feature distribution $h$.

In conclusion, existing algorithms can be adapted straightforwardly by adding an additional learning step, and evaluating the intrinsic reward function. This learning step can be integrated to on-policy algorithms, or to off-policy algorithms using solely transitions generated from the MDP when $N = 1$. In practice, we nevertheless observed that choosing the value of $N$ larger than one could drastically improve the learning process. The latter may require a slight adaptation to store multi-step transitions instead of one-step transitions. In Appendix B, off-policy actor-critic (Degris et al., 2012) and soft actor-critic (Haarnoja et al., 2018b) are adapted as advocated and used in experiments.

## 4 EXPERIMENTS

### 4.1 EXPERIMENTAL SETTING

Illustrative experiments are performed on adapted environments from the Minigrid suite (Chevalier-Boisvert et al., 2023). In the latter, an agent must travel across a grid containing walls and passages in order to reach a goal. The size of the grid and the number of passages and walls depend on the environment. The state space is composed of the agent's orientation, its position on the grid, as well as the positions of the passages in the walls and their orientations. In some environments, the goal to be reached is randomly generated and is also part of the state. The agent can take four different actions: turn left, turn right, move forward or stand still. The need for exploration comes from the sparsity of the reward function, which is zero everywhere and equals one in the state to be reached.

In the experiments, we asses the new MaxEntRL framework introduced in Section 3.1. In practice off-policy actor-critic (Degris et al., 2012), i.e., an approximate policy iteration algorithm, is adapted to the MaxEntRL framework as advocated in Section 3.3. This new algorithm is detailed in Appendix B and is called off-policy actor-critic with conditional visitation measures (OPAC+CV) in the remaining of the paper. For the Minigrid environments, the features $z \in \mathcal{Z}$ are the pairs of horizontal and vertical positions of the agent in the environment, the function $h$ is a deterministic mapping that computes these positions based on the state-action pairs, and the relative measure $q^*$ is uniform. The state and actions space representations, additional details about the function approximators, and other hyperparameters are provided in Appendix C.

The new MaxEntRL algorithm is compared to two alternative algorithms. The first concurrent method is soft actor-critic (SAC) (Haarnoja et al., 2018b). The latter is a commonly-used Max-EntRL algorithm where the feature space is the action space $\mathcal{Z} = \mathcal{A}$, the conditional distribution is the policy $q^\pi(z|s, a) = \pi(z|s)$ for all $a$, and the relative measure $q^*$ is uniform. To the best of our knowledge, the MaxEntRL framework used in soft actor-critic is also the unique alternative framework where policies can eventually be computed off-policy when the state and action space is large or continuous. The second concurrent method is a combination between off-policy actor-critic (Degris et al., 2012), and the intrinsic reward function from Lee et al. (2019) and Zhang et al. (2021a). We refer to that algorithm as off-policy actor-critic with marginal visitation measures (OPAC+MV). Here, the feature space $\mathcal{Z}$ is the same as in OPAC+CV, the conditional distribution $q^\pi(z|s, a)$ is the discounted visitation measure of features for all state $s$ and action $a$, and the relative measure $q^*$ is uniform. In practice, the state visitation measure is computed by maximum likelihood estimation (Lee et al., 2019), and the feature probability and intrinsic reward is computed as for OPAC+CV; more details are available in Appendix B. This algorithm is on-policy.

### 4.2 EXPLORING SPARSE-REWARD ENVIRONMENTS

The feature space coverage of optimal exploration policies computed with OPAC+CV and OPAC+MV are first compared. In Figure 1, the evolution of the entropy of the discounted visitation measure of features is represented as a function of the number of algorithm iterations, when only the intrinsic rewards are considered. For each environment, the entropy increases rapidly, and a high-entropy policy results from the optimization with both algorithms. For the environments `Empty-16x16` and `FourRooms`, OPAC+CV outperforms the concurrent method from far. In addition, OPAC+CV has smaller confidence intervals and less oscillations compared to the concurrent method. Our method is thus arguably more stable. We observed that action entropy regularization

made OPAC+MV more stable. It nevertheless comes at the cost of lower state entropy. It is worth noticing that while OPAC+CV does not stricto sensus optimize the discounted visitation measure, it performs at least equivalently to the concurrent method that does optimize this objective. In the literature, feature exploration is usually used to compute optimal exploration policies as an initialization when extrinsic rewards are not available. Our method is a robust off-policy alternative to traditional approaches. By way of illustration, the discounted visitation distribution of features is shown in Figure 2 for a particular instance of a Minigrid environment after optimization with OPAC+CV.

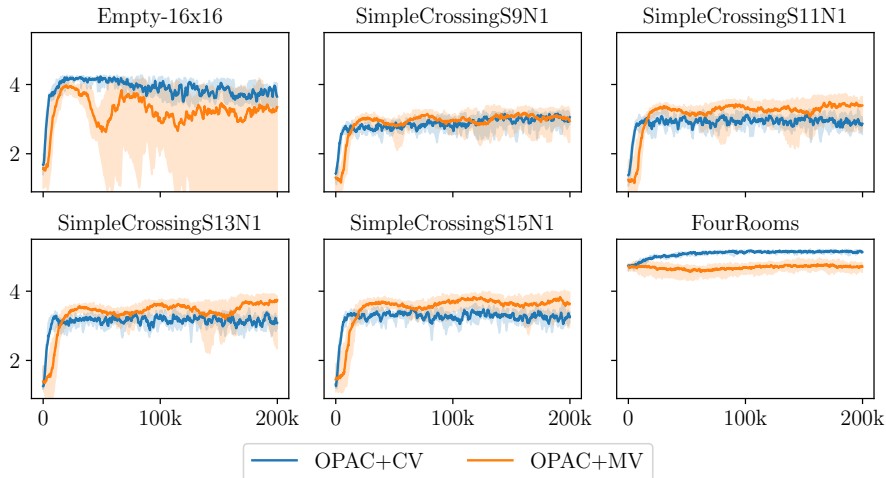

Figure 1: Evolution of the entropy of the discounted visitation probability measure of the position of the agent on the grid when computing exploration policies (i.e., when neglecting the rewards of the MDP). The entropy is computed empirically with Monte Carlo simulations. For each iteration, the median over five runs is reported, along with the highest and lowest values over these runs.

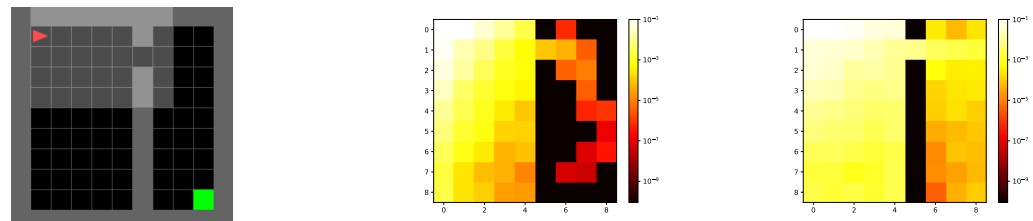

Figure 2: Discounted visitation probability measure of the position of the agent on the grid in the `SimpleCrossingS11N1`-environment. The initial position of the agent, with the environment composition, and the target are first represented. Second, the probability measure after the neural networks initialization is represented (middle figure) along with the probability measure after the policy optimization (right figure). The distribution is estimate using Monte Carlo simulations.

## 4.3 CONTROLLING SPARSE-REWARD ENVIRONMENTS

The objective of MaxEntRL is to provide intrinsic motivations to exploration in order to compute a high-performance policy. In Figure 3, the expected return of OPAC+CV and OPAC+MV is compared to the expected return of SAC, the most commonly used off-policy MaxEntRL algorithm. As can be seen, our method always performs at least as well as SAC. In the `SimpleCrossing`-environments, the two methods perform equivalently for the first one, OPAC+CV performs similarly to the lucky realizations of SAC for the second one, and only OPAC+CV computes policies with non-zero return for the last two. These environments, one being illustrated in Figure 2, are open grids of different sizes where the agent shall cross a wall through a small passage to reach the target. The larger the environment, the lower the probability of reaching the goal with a uniform

policy, and the worst the performance of SAC. The same can be observed in the `Empty-16x16`-environment. On the contrary, both MaxEntRL methods perform equivalently in the `FourRooms`-environment, where complex exploration is apparently not necessary to solve the problem. Finally, OPAC+CV and OPAC+MV perform similarly. In the environments `SimpleCrossingS13N1` and `SimpleCrossingS15N1`, the concurrent method outperforms OPAC+CV. This is mostly due to the difference in reward scales of both methods, and the absence of scheduling on the intrinsic reward weight $\lambda$. Probably the most important is that both methods allow to compute policies with non-zero rewards. With an appropriate scheduling on the intrinsic reward weighting, both methods could eventually compute high-performing policies.

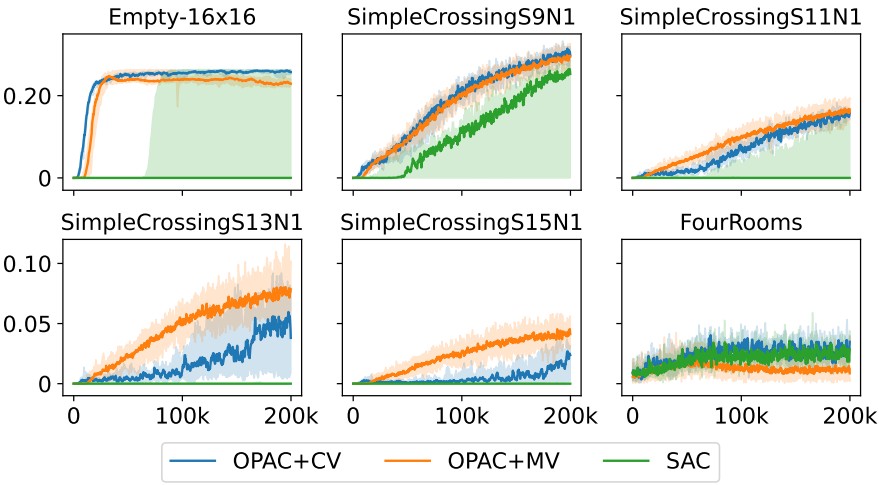

Figure 3: Expected return during the policy optimization with OPAC+CV, OPAC+MV, and SAC. The expectation is computed empirically with Monte Carlo simulations. For each iteration, the median over five runs is reported, along with the highest and lowest values over these runs.

## 5 CONCLUSION

In this paper, we presented a new MaxEntRL framework providing intrinsic reward bonuses proportional to the entropy of the distribution of features built from the states and actions visited by the agent in future time steps. The reward bonus can be estimated efficiently by sampling from the conditional distribution of states visited, which we proved to be the fixed point of a contraction mapping and can be learned for any policy relying on batches of arbitrary transitions. In this MaxEntRL framework, we propose the first end-to-end off-policy algorithm that allows to effectively explore the state and action spaces. The algorithm is benchmarked on several control problems. The method we developed is easy to implement, works with a large range of parameters across many environments and can be integrated into already existing RL algorithms.

Future works include testing and adapting the algorithm to continuous state action spaces, which can straightforwardly be done using continuous neural density estimators like normalizing flows. Furthermore, in this paper, the feature space to explore is fixed a priori, but could be learned. A potential avenue is to explore reward-predictive feature spaces. Finally, the distribution that is learned for exploration purpose can be used to generate new samples to enhance the sample efficiency when learning the critic. The integration of the latter into the MaxEntRL framework is left for future work.

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

## A    PROOFS THEOREMS

**Proof Theorem 1.**  Let us express the state-action value function as a function of the conditional state-action visitation distribution (Eysenbach et al., 2020)

$$
\begin{aligned}
Q^\pi(s,a) &= \int d^{\pi,\gamma}(s_\infty, a_\infty | s, a) R(s_\infty, a_\infty) \, ds_\infty \, da_\infty \\
&= \int \frac{d^{\pi,\gamma}(s_\infty, a_\infty | s, a)}{d^{\pi^*,\gamma}(s_\infty, a_\infty | s, a)} d^{\pi^*,\gamma}(s_\infty, a_\infty | s, a) R(s_\infty, a_\infty) \, ds_\infty \, da_\infty \\
&\geq Q^{\pi^*}(s,a) \inf_{s_\infty, a_\infty} \frac{d^{\pi,\gamma}(s_\infty, a_\infty | s, a)}{d^{\pi^*,\gamma}(s_\infty, a_\infty | s, a)} \quad\quad (17) \\
&= Q^{\pi^*}(s,a) \exp \inf_{s_\infty, a_\infty} \left( \log \frac{d^{\pi,\gamma}(s_\infty, a_\infty | s, a)}{d^{\pi^*,\gamma}(s_\infty, a_\infty | s, a)} \right) \\
&= Q^{\pi^*}(s,a) \exp \left( \inf_{s_\infty, a_\infty} \left( \log d^{\pi,\gamma}(s_\infty, a_\infty | s, a) - \log d^{\pi^*,\gamma}(s_\infty, a_\infty | s, a) \right) \right) \\
&= Q^{\pi^*}(s,a) \exp \left( - \sup_{s_\infty, a_\infty} \left( \log d^{\pi^*,\gamma}(s_\infty, a_\infty | s, a) - \log d^{\pi,\gamma}(s_\infty, a_\infty | s, a) \right) \right) \\
&\geq Q^{\pi^*}(s,a) \exp \left( - \sup_{s_\infty, a_\infty} \left| \log d^{\pi^*,\gamma}(s_\infty, a_\infty | s, a) - \log d^{\pi,\gamma}(s_\infty, a_\infty | s, a) \right| \right) \quad (18) \\
&= Q^{\pi^*}(s,a) \exp \left( - \| \log d^{\pi^*,\gamma}(\cdot, \cdot | s, a) - \log d^{\pi,\gamma}(\cdot, \cdot | s, a) \|_\infty \right) \ .
\end{aligned}
$$

Equation (17) holds by the monotonicity of the (Lebesgue) integral, equation (18) holds as $\sup_x f(x) \leq \sup |f(x)|$ for any function $f$. Finally, reorganizing the expression, we obtain the statement from Theorem 1. It generalizes the results from Kakade & Langford (2002).

$\square$

**Proof Theorem 2.**  For all conditional distributions $p$ and $q$

$$
\begin{aligned}
\sup_{s,a} L_n(\mathcal{T}^\pi p(\cdot|s,a), \mathcal{T}^\pi q(\cdot|s,a))^n &= \sup_{s,a} \int \| \mathcal{T}^\pi p(s_\infty|s,a) - \mathcal{T}^\pi q(s_\infty|s,a) \|_n \, ds_\infty \\
&= \gamma \sup_{s,a} \int \left\| \mathbb{E}_{\substack{s' \sim p_1(\cdot|s,a) \\ a' \sim \pi(\cdot|s')}} [p(s_\infty|s',a') - q(s_\infty|s',a')] \right\|_n \, ds_\infty \\
&\leq \gamma \sup_{s,a} \int \mathbb{E}_{\substack{s' \sim p_1(\cdot|s,a) \\ a' \sim \pi(\cdot|s')}} [\| p(s_\infty|s',a') - q(s_\infty|s',a') \|_n] \, ds_\infty \\
&= \gamma \sup_{s,a} \mathbb{E}_{\substack{s' \sim p_1(\cdot|s,a) \\ a' \sim \pi(\cdot|s')}} \left[ \int \| p(s_\infty|s',a') - q(s_\infty|s',a') \|_n \, ds_\infty \right] \\
&\leq \gamma \sup_{s,a} \sup_{s',a'} \left( \int \| p(s_\infty|s',a') - q(s_\infty|s',a') \|_n \, ds_\infty \right) \\
&= \gamma \sup_{s',a'} \int \| p(s_\infty|s',a') - q(s_\infty|s',a') \|_n \, ds_\infty \\
&= \gamma \sup_{s,a} L_n(p(\cdot|s,a), q(\cdot|s,a))^n
\end{aligned}
$$

$\square$

## B   Soft Actor-Critic and Off-Policy Actor-Critic with Conditional visitation measure

In the following, we adapt soft actor-critic (Haarnoja et al., 2018b), itself an adaptation of off-policy actor-critic (Degris et al., 2012), according to the procedure from Section 3.3. In essence, soft actor-critic estimates the state-action value function with a parameterized critic $Q_\phi$, which is learned using expected SARSA (sometimes called generalized SARSA), and updates the parameterized policy $\pi_\theta$ with approximate policy iteration (i.e., off-policy policy gradient), all based on one-step transitions stored in a replay buffer $\mathcal{D}$. The actor and critic loss functions are furthermore extended with the log-likelihood of actions weighted by the parameter $\lambda_{SAC}$, therefore called soft and considered a MaxEntRL algorithm using the entropy of policies as intrinsic reward. In the particular case where $\lambda$ equals zero, the algorithm boils down to a slightly revisited implementation of off-policy actor-critic.

Soft actor-critic is adapted to MaxEntRL with the intrinsic reward function defined in Section 3.1, as follows. First, $N$-step transitions are stored in the buffer $\mathcal{D}$ instead of one-step transitions. Second, the conditional state visitation distribution is estimated with a function approximator $d_\psi$ and learned with stochastic gradient descent applied on the loss function defined in equation (15). Third, at each iteration of the critic updates, the reward provided by the MDP is extended with the intrinsic reward.

Formally, the parameterized critic $Q_\phi$ is iteratively updated performing stochastic gradient descent steps on the loss function

$$\mathcal{L}(\phi) = \mathop{\mathbb{E}}_{s_t, a_t \sim \mathcal{D}} \left[ \left( Q_\phi(s_t, a_t) - y \right)^2 \right] \tag{19}$$

$$y = R(s_t, a_t) + \lambda R^{int}(s_t, a_t) + \gamma \left( Q_{\phi'}(s_{t+1}, a_{t+1'}) - \lambda_{SAC} \log \pi_\theta(a_{t+1'}|s_{t+1}) \right) , \tag{20}$$

where $a_{t+1'} \sim \pi_\theta(\cdot|s_{t+1})$, and where $\phi'$ is the target network parameter.

Furthermore, the policy $\pi_\theta$ is updated performing gradient descent steps on the loss function

$$\mathcal{L}(\theta) = - \mathop{\mathbb{E}}_{s_t, a_t \sim \mathcal{D}} \left[ \log \pi_\theta(a_{t'}|s_t) A(s_t, a_{t'}) \right] \tag{21}$$

$$A(s_t, a_{t'}) = Q_\phi(s_t, a_{t'}) - \lambda_{SAC} \log \pi_\theta(a_{t'}|s_t) , \tag{22}$$

where $a_{t'} \sim \pi_\theta(\cdot|s_t)$.

Algorithm 1 summarizes the learning steps during each iteration[1]. It differs slightly from the original soft actor-critic (Haarnoja et al., 2018b). The loss equation (21) is based on the log-trick instead of the reparametrization trick, the expected SARSA update in equation (19) is approximated by sampling, and a single value function is learned, as implemented in CleanRL (Huang et al., 2022). These changes are of minor importance in our experiments.

---

[1]A GitHub repository will be made public after the blind review process.

**Algorithm 1** SAC with conditional visitation measure for exploration

---

Initialize the policy $\pi_\theta$, the soft critic $Q_\phi$, and the visitation model $d_\psi$
Initialize the critic target $Q_{\phi'}$ and visitation target $d_{\psi'}$
Initialize the replay buffer with random $N$-step transitions
**while** Learning **do**
    Sample transitions from the policy $\pi_\theta$ and add them to the buffer
    **while** Update the visitation model **do**
        Sample a batch of $N$-step transitions from the buffer
        Perform a stochastic gradient descent step on $\mathcal{L}(\psi)$
    **end while**
    **while** Update the critic **do**
        Sample a batch of $N$-step transitions from the buffer (use only the 1-step transitions)
        For each element of the batch sample $z_t \sim q^\pi(\cdot|s_t, a_t)$
        Estimate the intrinsic reward $R^{int}(s_t, a_t) = \log q^*(z_t) - \log q^\pi(z_t|s_t, a_t)$
        Perform a stochastic gradient descent step on $\mathcal{L}(\phi)$
    **end while**
    Sample a batch of $N$-step transitions from the buffer (use only the 1-step transitions)
    Perform a stochastic gradient descent step on $\mathcal{L}(\theta)$
    Update the target parameters with Poliak averaging
**end while**

---

## C HYPERPARAMETERS EXPERIMENTS

In this section, we detail implementation details for reproducing the experiments. In practice, the agent observes the concatenation of the one-hot-encoding of the components of the state space and takes actions in one-hot-encoding format too. The policy $\pi_\theta$ is a neural network that outputs a categorical distribution over the action representation. The critic $Q_\phi$ is a neural network that takes as input the concatenation of the state and action representations and outputs a scalar. In OPAC+CV, the visitation distribution model $d_\psi$ is also a neural network that takes the same input as the critic $Q_\phi$ and outputs, for each component of the state space, a categorical distribution over its one-hot-encoding representation. In OPAC+MV, the visitation distribution model $d_\psi$ is a marginal distribution over the same one-hot-encoding representation. In both algorithms, this amounts to assuming the conditional independence of the future state components given the state and action taken as input. This implementation choice mitigates the curse of dimensionality. In addition, it allows to compute the probability of a feature in closed form. The probability equals the product of the probability of the vertical position and the probability of horizontal position provided in one-hot-encoding by the model $d_\psi$. Table 1 summarizes the hyperparameters used in the experiments. In practice, two different discount factors were used, and the parameter $\lambda_{SAC}$ had no significant influence in the different experiments and was kept constant for SAC, OPAC+CV, and OPAC+MV simulations.

Table 1: Hyperparameters

| Parameter | Value |
|---|---|
| Neurons for each network layers | 256 |
| Layers policy | 2 |
| Layers critic | 2 |
| Learning rate policy | 0.00001 |
| Learning rate critic | 0.0001 |
| Maximum trajectory length | 200 |
| Buffer size | 1000 |
| Batch size | 32 |
| Target update weight $\tau$ | 0.1 |
| Control $\gamma$ | 0.95 |
| SAC $\lambda_{SAC}$ | 0.0001 |
| Layers visitation model OPAC+CV | 2 |
| Learning rate visitation model | 0.00001 |
| MaxEntRL $\lambda$ | 0.01 |
| Target update weight $\tau$ | 1 |
| Visitation $\gamma$ | 0.9 |
| $N$ | 5 |

