# OpenReview forum: "Off-Policy Maximum Entropy RL with Visitation Measures"
_ICLR.cc/2025/Conference — Submitted to ICLR 2025_

### Official Review · Reviewer_MQXP · 2024-11-02

**Soundness:** 1
**Presentation:** 3
**Contribution:** 2
**Rating:** 5
**Confidence:** 4

**Summary:**

The authors propose a maximum entropy RL algorithm whose optimization objective is composed of from two reward components. One is the classic reward function and the other is an intrinsic reward, added for improving exploration. This second term is inversely proportional to the divergence with respect to some high-entropy distribution, such as the uniform. Specifically, the authors propose to minimize the discounted occupancy measure of the policy towards this target distribution. Since the discounted occupancy measure is unknwon, an approximation of this distribution is also optimized durig the main loop. They develop an adaptation of Soft Actor Critic (SAC) for a surrogate of their objective function and they conducted some validation experiments in discrete domains.

**Strengths:**

The paper addresses a very relevant topic for the RL comminity. There are many works in maximum-entropy RL and it is true that, unlike this work, most papers focus on maximizing the entropy of the immediate actions. Maximizing the visitation frequency is a recent algoritmic addition, which deserves further study.

The proposed method is relatively easy to implement. In particular, eq (12) appears to be a practical approximation of the objective that was originally considered for the discounted occupancy measure.
The paper is quite easy to follow and presentation quality is good.

The authors claim that the source code of the algorithm will be made public after acceptance.

**Weaknesses:**

I believe that the two theorems are not strong enough to support some important claims of from the paper.
1. [Theorem 1] In the abstract, the authors say that maximizing the intrinsic rewards is related to minimizing an approximation of the suboptimality gap, thus, suggesting that a suboptimality theorem proves the correctness of the proposed intrinsic rewards. However, by looking at the proof Theorem 1, we can see that the statement mostly comes from an algebraic manipulation of the terms, and the same inequality would also hold for any couple of policies $\pi$, $\pi^*$, and for any third term $q^*$. Therefore, this does not confirm the correctness of the intrinsic rewards, as previously implied in the paper. Lastly, regarding the tightness of the result, we can see that arbitrary small differences near 0 probability become maximally wide suboptimality gaps, making the inequality trivially satisfied for common policies.
2. [Theorem 2] Theorem 2 also lack specificity with respect to the proposed algorithm. Indeed, it only proves that the Bellman update for the discounted occupancy measure is contractive, which I would regard as a widely known fact. This is due to the tight relationship between each component of unnormalized occupancy measure and the value function with respect to a positive reward in each state component.

Regarding the experimental evaluation:

3. [Experiments] The experiments test the proposed algorithm against one baseline, SAC, in a few gridworld environments. From my experience, in such small environments, I would also expect SAC to converge, even if at a slower rate than the competitor. Instead this is not the case, and in some environments, SAC does not sample the goal even once. Unfortunately, I cannot exclude that other factors may have prevented SAC from a more natural convergence. For example, has $\alpha$ been tuned in the version of the algorithm that does not use $\lambda$? The limited set of environments and algorithms considered is also a limitation of the experimental evaluation.

Minor:
- The role of the function $g()$ is unclear. In the text, the authors say that this can be an "arbitrary distribution over the state". However, different functions, such as deterministic one, would dramatically alter the objectives.
- $\pi^*$ in Theorem 1 has not been defined. This might be misunderstood among: the optimum with respect to the original rewards, the intrinsic rewards, or the combined rewards.
- The norm appearing in Theorem 2 has not been defined.

**Questions:**

The authors may address any of the weaknesses discussed above.

---

> ### Author Response · Authors · 2024-11-26
> **Response to reviewer MQXP with comments addressed in the revised paper**
>
> Dear reviewer, thank you for your proofreading and comments. The weaknesses you pointed out have been addressed in the revised version of the paper. We believe that the paper is now in line with ICLR standards for acceptance and kindly ask you to change your score accordingly.
>
> > Theorem 1.
>
> Theorem 1 justifies the use of our new intrinsic reward function. For a fixed arbitrary target $q^*$, an optimal exploration policy is one that minimizes the bound in Theorem 1, under some assumptions. Please refer to the revised discussion in Section 3.1, where this point is clarified.
>
> The case where $\pi$ and $\pi^\*$ are fixed to afterwards find the best third term $q^*$ is irrelevant to the optimization of optimal exploration policies, where the target is fixed. Please refer to the revised discussion in Section 3.1.
>
> Assuming the target distribution and conditional discounted state occupancy measure are smooth and nonzero everywhere, the bound is never satisfied for trivial cases. Please refer to the revised discussion in Section 3.1, where assumptions have been clarified.
>
> > Theorem 2.
>
> Theorem 2 guarantees that there exists a unique conditional state distribution and that it can be learned under some assumptions. To the best of our knowledge, we are first to prove this property. If the reviewer provides a reference where the property is already proven, we can remove the claim stating we are first to show this theorem.
>
> > Experiments.
>
> First, we have extended the experimental setting to include a fair comparison with other MaxEntRL algorithms. Please refer to Section 4 in the revised manuscript. In practice, the MiniGrid environments are pretty challenging for exploration. Our largest environment has $4 \times 15 \times 15 \times 15 \times 2=27000$ different possible discrete states.
>
> In practice, different values for the factor $\alpha$ (which we have renamed in the updated version) were tested and it has no large influence in sparse environments. As long as zero-rewards are observed, the exploration bonus dominates and the algorithm tends to make the actions uniformly distributed. Once a non-zero reward is observed, SAC converges towards an optimal policy rapidly. In practice, this will eventually always happen, but, in large environments as those we consider, the probability to observe a non-zero reward is minuscule.
>
> > Minor.
>
> 1. The distribution $g$ will indeed in practice influence the algorithm. It is the same consideration as in most RL algorithms where we want to optimize an objective for all state and action, but eventually solve it on expectation over some state-action pairs. A typical example is (approximate) policy iteration. To guarantee convergence, it is then necessary to assume non-zero probability over each pair. We did not enter into these considerations, as we do not provide theoretical guarantees for convergence. In practice, the function $g$ is the distribution of state and action pairs in the replay buffer. This has been clarified in Section 3.3.
> 2. The optimal policy is always the policy maximizing the rewards of the MDP only, as defined in section 2.1.
> 3. The norm has been explicitly given in the revised version, see Theorem 2.

---

> ### Author Response · Authors · 2024-11-26
> **Follow-up Theorem 1**
>
> Dear reviewer, we have updated Theorem 1 and the relative paragraph to simplify the interpretation, as requested by reviewer 6YUn. Please refer to the updated manuscript. It should also furthermore clarify some of the points you raised yourself. We believe that the paper is in-line with the ICLR standard for acceptance, and we once again ask you to revise your score for acceptance.

---

> > ### Comment · Reviewer_MQXP · 2024-12-01
> >
> > I thank you the reviewers for the updated manuscript and the replies.
> >
> > *Theorem 1*. The meaning of Theorem 1 is clear and its proof seems to be correct. Unfortunately, my biggest concerns remain:
> > It is not reasonable to assume that the occupancy of $\pi^*$ is strictly positive and bounded from below in any state. When this does not happen, the bound becomes trivially true. More in general, for a wide range of low occupancy measures this bound is also trivially satisfied. Moreover, Theorem 1 plays the same role as the equality $V^* - V^\pi = \langle d^* - d^\pi, R \rangle / (1-\gamma)$, which tight and widely known.
> >
> > *Theorem 2*. To my understanding, this theorem does not talk about the how the occupancy might be learnt from data. It only shows that the Bellman operator for the occupancy distribution is contractive. Regarding previous results about contractivness property, the authors may refer to the properties of the "Bellman flow operator". One possibile reference is Fan and Ramadge 2020 A Contraction Approach to Model-based Reinforcement Learning, although I suspect that previous references exists under different names.
> >
> > Given the above, I am not willing to increase my score.

---

> > > ### Author Response · Authors · 2024-12-01
> > > **Adressing last concerns**
> > >
> > > Thank you for your response. Below, we provide detailed answers to your remarks.
> > >
> > > 1. The theorem holds for any conditional occupancy measure, and there is no requirement for the occupancy to be bounded below in all states. The equation you reference is indeed tight and can be adapted to account for the Q-function instead of the V-function. However, it does not provide a bound on the Q-function in terms of the distance between conditional occupancy measures, as Theorem 1 does.
> > > 2. Theorem 2 establishes that the operator is contractive, meaning the conditional occupancy measure is its unique fixed point. This measure can therefore be learned either through successive application of the operator to any conditional distribution or by minimizing the residual (i.e., the distance between $\mathcal{T}^\pi d$ and $d$). In practice, since the residual cannot be computed with the $\bar L_n$-norm, we minimize the cross-entropy (that can be seen as an approximation of the KL metric). With this approach, the loss can be estimated based on samples, enabling practical computation of the distribution. This method also generalizes algorithms described in the introduction (line 087). Additionally, the operator we use is distinct from the Bellman flow referenced in your comments. To the best of our knowledge, we are the first to prove the contractiveness of the operator we employ.
> > >
> > > These answers should allay your concerns. We believe that your comments do not invalidate our contribution: introducing a new MaxEntRL framework with a novel intrinsic reward for exploration and proposing a practical off-policy algorithm for learning policies that maximize this new objective.
> > >
> > > We kindly ask you to reconsider your score for acceptance.
> > >
> > > The Authors

---

> > > > ### Author Response · Authors · 2024-12-02
> > > >
> > > > Dear Reviewer,
> > > >
> > > > Thank you again for your feedback on our paper. As the discussion period ends, we kindly ask you to reconsider increasing your score if you believe that your concerns have been addressed.
> > > >
> > > > Best regards,
> > > >
> > > > The Authors

---

### Official Review · Reviewer_6YUn · 2024-11-03

**Soundness:** 1
**Presentation:** 2
**Contribution:** 3
**Rating:** 3
**Confidence:** 4

**Summary:**

In this paper, the authors propose a new framework for MaxEnt RL based on conditional state-action visitation distributions $d^{\pi,\gamma}(s',a'|s,a)$ (in contrast to non-conditional and/or state-based methods). Theoretically, the authors provide a suboptimality bound for using this framework and prove that a unique optimal policy exists by reframing the method using Bellman operators. Empirically, the authors show their method achieves good coverage on a number of benchmarks.
Next, the authors show how their method can be incorporated in off-the-shelf RL algorithms to provide online of off-line policy improvements. Empirically, they show this improves the performance of SAC on the same benchmarks.

**Strengths:**

The paper's topic is interesting and significant: entropy maximization is a promising direction for improving exploration, which can often be a bottleneck in RL. The paper nicely combines both theory and application. Moreover, the proposed method can easily be incorporated into existing methods, which makes it easier to apply in practice.

**Weaknesses:**

In my opinion, this paper suffers from three main weaknesses:

**1: The comparison to prior works is lacking.** The main contribution of this paper is the introduction and analysis of a framework that uses conditional state-action visitation distributions $d^{\pi,\gamma}(s',a'|s,a)$. However, no intuition is given as to why this is the right choice. Moreover, the authors do not cite or compare to [1], which uses the (very similar) non-conditional state-action visitation $d^{\pi,\gamma}(s',a')$ instead. A more in-depth discussion of the pros and cons of the proposed method, as well as a more thorough explanation of existing work, is required.

**2: The empirical evaluation is limited.** In the experiments, the proposed method is only compared to SAC. However, SAC uses entropy regularization for its actions, not the states it reaches, and thus, it is unsurprising that it performs worse on the chosen benchmarks. No comparison is made to other methods that use state visitation distributions. Without such a comparison, I find it hard to judge whether the proposed method improves prior works. Moreover, the experiments do not give any information on the additional computational cost of the method: these should be mentioned.

**3: The proven suboptimality gap (Theorem 1) is confusing.** In the theoretical analysis, the authors show a bound for the sub-optimality of a policy based on the conditional state-action distribution. However, the implications of this bound are poorly explained. As far as I understand, the gap cannot practically be computed since it depends on the (unknown) visitation distribution of an optimal policy. Furthermore, the choice of $q^*$ is arbitrary: since the proof uses a triangle inequality (in equation 17, Appendix A), $q^*$ could be replaced by any function to give a valid bound, and there is no reason to assume $q^*$ would yield a tighter bound than any other function. Lastly, a policy that optimizes this bound does not necessarily perform better than one that does not, so we cannot use it for policy improvement. Thus, I don't understand why the bound is provided: it should either be removed, or the implications should be explained more clearly.

[1] : Zhang et. al., Exploration by Maximizing Renyi Entropy for Reward-Free RL Framework. AAAI, 2021.

**Questions:**

I've already outlined my questions in the 'weaknesses' section above. In particular:
1)  how does the proposed method compare to prior works, both intuitively and empirically?
2) What conclusions can we draw from the suboptimality gap in Theorem 1?

---

> ### Author Response · Authors · 2024-11-26
> **Response to reviewer 6YUn with comments addressed in the revised paper**
>
> Dear reviewer, thank you for your proofreading and comments. The weaknesses you pointed out have been addressed in the revised version of the paper and the questions you asked have also been answered. We believe that the paper is now in line with ICLR standards for acceptance and kindly ask you to change your score accordingly.
>
> > The comparison to prior works is lacking.
>
> We have clarified the main contribution in the introduction, which is the following: we introduce a new MaxEntRL framework, i.e., a new intrinsic reward for exploration, and propose a practical algorithm for learning policies that maximize this new MaxEntRL objective. The introduction has also been revised to provide a better overview of the literature, the limitations of existing algorithms, and better set the scope of our contribution. The revised experimental setting now includes a concurrent method similar to the algorithm developed by Zhang et. al.
>
> The work by Zhang et. al. has been included in the introduction and is indeed part of many algorithms that optimize the entropy of the non-conditional state-action visitation. We can distinguish three clases of such algorithms. First, there exist model-based algorithms using a tabular representation of the transition matrix and the initial distribution. Some are off-policy and have convergence guarantees, but are limited to (small) discrete spaces. There exist algorithms that estimate the density with kNN, they are on-policy. Finally, other algorithms use neural density estimators, they are arguably the best choice when large and/or continuous state and actions spaces are at hand. They are nevertheless also on-policy. We introduce a new intrinsic reward that can be estimated using a neural density estimator, which is learned off-policy using arbitrary transitions. Furthermore, the (revised) experimental results are concurrent with the existing methods.
>
> Our framework combines several advantages:
> * The MaxEntRL objective promotes (discounted) infinite-time exploration of a feature space.
> * The intrinsic reward can be estimated with a parametric function (a neural density estimator like a categorical distribution for discrete spaces, or a normalizing flow in general), and this function can be learned off-policy.
> * The learning procedure and intrinsic reward estimation is less subject to the exponential decreasing influence of future states compared to exploration with the marginal discounted state visitation measure.
>
> The claims have been adjuster to set the scope on this new reward function, and our algorithm has been benchmarked against alternative methods from the literature. The title of the paper has also been modified to better distinguish our MaxEntRL framework with existing ones.
>
> > The empirical evaluation is limited.
>
> We have included a new experiment where MaxEntRL with the (non-conditional) discounted state visitation is confronted to our method. Results highlight advantages of our method, which is more stable and is off-policy. Section 4 has been revised, and an extensive discussion was added.
>
> We have not observed any significat difference in computational cost between the methods we compared.
>
> > The proven suboptimality gap (Theorem 1) is confusing.
>
> Theorem 1 justifies the use of our new intrinsic reward fucntion. Under some assumptions, an optimal exploration policy is one that minimizes the bound in Theorem 1. The target $q^*$ is arbitrary, and corresponds to the relative entropy measure in the MaxEntRL framework. The discussion has been updated in the revised paper to clarify this point.
>
> > how does the proposed method compare to prior works, both intuitively and empirically?
>
> The introduction has been updated to better compare our approach to prior works, and Section 4 includes additional experiments in the revised paper. See responses to previous questions, and please refer to the revised introduction and Section 4.
>
> > What conclusions can we draw from the suboptimality gap in Theorem 1?
>
> Theorem 1 justifies the use of our new intrinsic reward function. Under some assumptions, an optimal exploration policy is one that minimizes the bound in Theorem 1. The discussion has been updated in the revised Section 3.1 to clarify this point.

---

> > ### Comment · Reviewer_6YUn · 2024-11-26
> > **Follow-up**
> >
> > Thanks to the authors for their in-depth answers. I have two follow-up questions:
> >
> > 1) *Contributions:* After your revision of the experimental section, it seems like the main strength of your work is that it is off-policy, as opposed to prior methods. However, it seems to me that off-policy methods do not need to use the specific visitation measure you propose (Eq 4). Is this true, and if so, what is the advantage of using this visitation measure specifically?
> > 2) *Theorem 1:* Despite the revision and comment about Thm 1, I am still unsure as to what you are trying to show.  Let me try to be more concrete. From your comment and the revised test, you seem to suggest that given a $q^*$, a valid approach to find a good exploratory policy is to minimize the given bound. However, this is often nonsensical. For example, consider picking $q^*$ as a Dirac distribution for some state-action pair. Then, making our visitation density match this distribution minimizes the bound but clearly does not give good exploratory behavior. Thus, why would this be a good approach? If it is not, what is Thm 1 trying to show instead?

---

> > > ### Author Response · Authors · 2024-11-26
> > > **Response follow-up**
> > >
> > > Thanks you for your follow-up. We hope that the following elements will respond to your questions.
> > >
> > > > Contributions
> > >
> > > One of the main advantages of our method is indeed that it allows to compute the intrinsic reward off-policy. Other methods exploring the state space are only off-policy in one particular case. In discrete state and action spaces, you can learn a model of the transition matrix of the MDP and of the initial distribution using samples. There is then a closed-form expression of the marginal distribution of states, which can be used to compute the intrinsic reward. The resulting algorithm is off-policy. This approach is limited to (small) discrete state and action spaces in tabular model-based RL, and requires to inverse the estimated transition matrix, which may in addition require a large number of samples to have an accurate estimation of the marginal state distribution.
> > >
> > > > Theorem 1.
> > >
> > > Theorem 1 should be used to interpret the quality of an optimal exploration policy, i.e., a policy maximizing the expected return computed only with the intrinsic reward we introduced. We prove, under some assumptions, that this policy minimizes the bound introduced in Theorem 1. The choice of $q^*$ corresponds to the choice made a priori when defining the exploration objective. Should the exploration strategy visit uniformly the state (or feature) space, visit it according to a gaussian distribution? This choice influences the quality of the exploration policy, and thus, according to Theorem 1, the return of that policy. Choosing to explore only one state (which corresponds to the diract example you used) is likely a very bad choice in many situations, and will indeed result in a bad exploration policy. In conclusion, we agree that the choice of $q^*$ is non-trivial, but it is not particular to Theorem 1, but to the choice of how to explore in general.
> > >
> > > If this clarifies your questions, we will appreciate if you reconsider the score of the paper, as we believe it is now in-line with the ICLR acceptance criteria.

---

> > > > ### Comment · Reviewer_6YUn · 2024-11-26
> > > >
> > > > Thanks to the authors for their quick response. However, I do not feel like my concern about Thm 1 has been sufficiently addressed.
> > > >
> > > > You mention that '$q^*$ corresponds to the choice made a priori when defining the exploration objective". However, as pointed out in my original review (as well as by Reviewer MQXP), the choice of $q^*$ in the theorem is arbitrary. The theorem would hold for any term (including ones that have nothing to do with the target distribution), and there is no reason given as to why $q^*$ is picked. Thus, I still see no underlying connection between the chosen framework and the bound, nor how it helps 'interpret the quality of an optimal exploration policy', and it is unclear to me what the theorem is trying to convey.

---

> > > > > ### Author Response · Authors · 2024-11-26
> > > > > **Follow-up Theorem 1**
> > > > >
> > > > > Thank you for the interaction and thank you for challenging the interpretation. Let us rephrase the point we were trying to make, and we look forward to your response.
> > > > >
> > > > > We have introduced a new objective for exploration, i.e., a new intrinsic reward. We see in the paper that this reward has some good learning properties (off-policy) and eventually allows us to challenge in practice traditional methods in two aspects. First, optimal exploration policies have a uniform state space (or feature space) coverage, which is usually considered as a standard exploration objective. Second, the control policies are high performing. The latter means that we do not sacrifice too much performance in exchange for good exploration, and, with a good reward weight scheduling, we will eventually have good final policies. We believe that the paper has at this point already proven to challenge the state of the art.
> > > > >
> > > > > We nevertheless want to give additional intuition about two points. What can we say about the quality of an optimal exploration policy? How is that quality dependent on the choice of target distribution?
> > > > >
> > > > > To answer the previous questions, we rely on Theorem 1. The latter is indeed valid for any target function $q^*$ and, in particular, for the target in the MaxEntRL framework we introduce. In that case, under some assumptions, there is furthermore an equivalence between the two objectives. The latter can be used to justify our framework.
> > > > >
> > > > > We acknowledge that we may have underestimated the complexity of our demonstration. We have therefore revised Theorem 1 and the relative explanation to clarify the interrogation raised by the reviewer(s). An updated manuscript has been uploaded. Please refer to the corresponding (colored) paragraph, page 4.
> > > > >
> > > > > We believe that the problem has been resolved and ask the reviewer to revise his score for acceptance. If the reviewer believes that the interpretation is still complex, we are willing to move it to an appendix, as it is not absolutely mandatory for defending the thesis of the paper.

---

> > > > > > ### Author Response · Authors · 2024-12-02
> > > > > >
> > > > > > Dear Reviewer,
> > > > > >
> > > > > > Thank you again for your feedback on our paper. As the discussion period ends, we kindly ask you to reconsider increasing your score if you believe that your concerns have been addressed.
> > > > > >
> > > > > > Best regards,
> > > > > >
> > > > > > The Authors

---

> > > > > > ### Comment · Reviewer_6YUn · 2024-12-02
> > > > > >
> > > > > > Thanks to the authors for the continued interactions. Although I think the revisions have improved the paper, I still think the implications of Theorem 1 are poorly explained. Let me make this more concrete using two lines from the paper. In line 218 you write the following:
> > > > > >
> > > > > > > Computing optimal exploration policies in the MaxEntRL framework we introduce can then be interpreted as a practical algorithm for computing a policy minimizing the upper bound in equation (8) provided a priori with the target.
> > > > > >
> > > > > > This is true, but this is only meaningful if minimizing this bound also yields good policies. However, we see that if $q^*$ does not closely resemble the visitation distribution of an optimal policy (as is the case in maximum entropy RL), then minimizing (8) will generally yield worse policies (see Eq. 7, which is a tighter bound). In other words, an optimal exploratory policy can (and often will) perform worse than a suboptimal one. Thus, the statement is correct, but it's implication (that the bound confirms the proposed framework yields good policies) is not.
> > > > > >
> > > > > > In the next line, you state the following:
> > > > > >
> > > > > > > The quality of the optimal exploration policy is then only dependent on this a priori choice of target.
> > > > > >
> > > > > > This is again true, but is only meaningful if your target distribution in some way tries to minimize the bound. However, as mentioned already, this is not the goal of this paper: it instead focuses on maximum entropy RL, which explicitly tries *not* to pick target distributions that match that of an optimal policy but instead focuses on exploring the entire environment. In other words, your choice of target distribution goes against what would minimize your bound. Thus, the statement is correct, but its implication (that the bound is used, or should be used, for determining a good target distribution) is not.
> > > > > >
> > > > > > In short, I do not think the theorem or your explanations are incorrect. However, I think they currently imply that there is some theoretical justification for your method that, as far as I can see, does not exist. I strongly believe that the implications of the theorem (specifically, on your framework) should be made more explicit, or the theorem should be removed.

---

> > > > > > > ### Author Response · Authors · 2024-12-03
> > > > > > >
> > > > > > > Dear Reviewer,
> > > > > > >
> > > > > > > Thank you for your response.
> > > > > > >
> > > > > > > Regarding your first comment, we agree. The optimality of the exploration policy depends on the target $q^*$, and it is possible for suboptimal policies to achieve a higher return. However, we did not claim that an optimal exploration policy is inherently good (i.e., near-optimal). Instead, we stated that the quality of the optimal policy is determined solely by the target distribution, as mentioned in line 220.
> > > > > > >
> > > > > > > For your second comment, you note that “maximum entropy RL tries not to pick target distributions that match that of an optimal.” This is only partially correct. The goal of MaxEntRL is to eventually compute near-optimal policies, as noted in line 040. Moreover, some works, such as Behavior from the Void by Liu and Abbeel, focus on the quality of exploration policy. The bounds we provide then indicate the expected return achievable with a given target $q^*$. You also mention that “the implication that the bound is used, or should be used, for determining a good target distribution is not correct.” We do not make this claim. The choice of $q^*$ can be motivated by various factors, and the optimal exploration policy’s quality can be bounded as we have shown.
> > > > > > >
> > > > > > > We propose to clarify two points:
> > > > > > > 1. The optimal exploration policy is not guaranteed to have a high return. It depends on the target distribution $q^*$.
> > > > > > > 2. In practice, the target $q^*$ is chosen without knowing the optimal occupancy and may intentionally differ depending on the application.
> > > > > > >
> > > > > > > We kindly ask you to consider these clarifications and also evaluate the rest of the paper in your final score.
> > > > > > >
> > > > > > > Best regards,
> > > > > > >
> > > > > > > The Authors

---

### Official Review · Reviewer_PF2o · 2024-11-03

**Soundness:** 3
**Presentation:** 3
**Contribution:** 1
**Rating:** 3
**Confidence:** 3

**Summary:**

The paper proposes a new intrinsic reward for exploration in RL, which combines the commonly used action entropy and another entropy term on the states visitation. Implementing this intrinsic reward requires to model the state visitation, here obtained through a TD-like methodology in a discounted setting. The intrinsic reward can be incorporated into any discounted RL algorithm, e.g., SAC, for improved exploration on the state space. The resulting method is evaluated against vanilla SAC on some sparse-rewards environment from MiniGrid.

**Strengths:**

- The paper introduces a general framework for MaxEnt RL encompassing all of the previous literature;
- The paper proposes an intrinsic reward that incentives entropy over states beyond the action entropy, which can benefit exploration in sparse-reward tasks especially;
- The paper makes an effort to motivate the resulting intrinsic reward through a theoretical result.

**Weaknesses:**

- The paper seems to be built on the misconception that entropy over states has not been thoroughly explored already, as it only mentions three related work on that stream. While the proposed method looks at least partially departing from previous works, this casts some clouds over the novelty of the contribution;
- The main theoretical result in Th. 1 is mostly implied by previous results, while the link with the resulting algorithm fades with the layers of approximations required by the practical implementation;
- The empirical analysis looks weak for an essentially methodological contribution. It only considers MiniGrid with discrete action space and not continuous control as much of the prior works. Moreover, the reported statistics are also non-standard and leaves the significance of the improvement unclear.

Given these concerns, which I further expand in the comments below, I am inclined to a negative evaluation of the paper. However, some of the technical solutions look interesting and putting the spotlight on what is new w.r.t. prior work may go a long way in making this paper a more substantial contribution to the field.

**COMMENTS**

1) Prior Work on State Entropy.

The paper mentions a few works on the maximization of the entropy of the visited states (Lee et al, Guo et al, Islam et al) but seems to neglect a large body of literature on this problem. The problem of the state entropy maximization has been first introduced by Hazan et al (2019) and then studied in a series of subsequent works (below). Notably, the works by Mutti et al (2021), Liu and Abbeel (2021), Seo et al (2021), Yarats et al (2021) also introduce an intrinsic reward derived from kNN entropy estimators that avoid to perform a costly density estimation. Even though the setting of this work may not perfectly match the discounted setting of this paper, I believe they shall be mentioned and discuss at grater length. I provide a (partial) list of the work in this stream below:
- Hazan et al., Provably efficient maximum entropy exploration, 2019;
- Mutti and Restelli, An intrinsically-motivated approach for learning highly exploring and fast mixing policies, 2020;
- Zhang et al., Exploration by maximizing Rényi entropy for reward-free rl framework, 2021;
- Mutti et al., Task-agnostic exploration via policy gradient of a non-parametric state entropy estimate, 2021;
- Liu and Abbeel, Behaviour from the void: Unsupervised active pre-training, 2021;
- Liu and Abbeel, Aps: Active pretraining with successor features, 2021;
- Seo et al., State entropy maximization with random encoders for efficient exploration, 2021;
- Yarats et al., Reinforcement learning with prototypical representations, 2021;
- Mutti et al., Unsupervised reinforcement learning in multiple environments, 2022;
- Mutti et al., The importance of non-Markovianity in maximum state entropy exploration, 2022;
- Nedergaard and Cook, k-means maximum entropy exploration, 2022;
- Mutti, Unsupervised reinforcement learning via state entropy maximization, 2023;
- Yang & Spaan, CEM: Constrained entropy maximization for task-agnostic safe exploration, 2023;
- Zisselman et al., Explore to generalize in zero-shot rl, 2023;
- Tiapkin, Fast rates for maximum entropy exploration, 2023;
- Jain et al., Maximum state entropy exploration using predecessor and successor representations, 2023;
- Kim et al., Accelerating reinforcement learning with value-conditional state entropy exploration, 203;
- Zamboni et al., How to explore with belief: State entropy maximization in pomdps, 2024;
- Zamboni et al., The limits of pure exploration in POMDPs: When the observation entropy is enough, 2024.

2) Novelty.

The previous comment also open some questions on the novelty of the presented intrinsic reward. In fairness, most of the previous works they are unsupervised (do not consider the presence of an external reward) and in the finite-horizon setting. However, extending them to discounted settings looks mostly doable and there are some papers studying the combination of external and intrinsic rewards (e.g., Seo et al. 2021). Especially, the claim for which this paper introduces the "first off-policy algorithm with state and action spaces exploration" seems a little overblown. RE3 also does that as it can be implemented with a SAC backbone. However, important implementation details between the two may make a huge difference

The main novelty seems to lie into the choice of a discounted visitation measure (instead of finite horizon) and an incentive for the entropy of future states (instead of all the states into the replay buffer). I believe the paper shall focus on those novel aspects and analyse how they contribute to improved performance.

Several time the method by Janner et al is mentioned in the paper, without an explicit comparison. Can the authors thoroughly compare their method with that of Janner et al?

3) Theoretical Ground for the Intrinsic Reward.

Result of Th. 1 looks a little bit trivial: It is basically saying that the error can be bounded with the infinity norm of the occupancy, which is well known (see Kakade and Langford, "Approximately optimal approximate reinforcement learning", 2002, and Pirotta et al., "Safe policy iteration", 2013). In practice, the problem would be to set a target $q^*$ that stays close to the occupancy of the optimal policy, which is non-trivial without prior knowledge.

Whereas the paper makes an effort to ground the method theoretically. However, going from the ideal method to the implemented version requires a series of changes and approximations that seems to diminish considerably the link with the theory.

4) Experiments.

The experiments are conducted with a quite severe discount (0.95 and 0.9 instead of the most common 0.99). However, it is unclear whether the sampling fo the trajectories is also discounted, or it merely considers chopped trajectories at a fixed step.

Figure 1: As mentioned above, other papers tried to maximize the entropy of the state distribution, why not comparing to them?

Figure 3: The choice of evaluation with the median, the min and max values is quite odd and a departure from common practices (Agarwalet al., "Deep RL at the edge of the statistical precipice", 2021). Moreover, even if the improvement over SAC turns out to be statistically significant, it warrants a comparison with other methods already trying to maximize the entropy of the states together with the entropy of the actions (e.g, RE3?)

5) Minor and Typos

In Theorem 2, the $\overline{L}_n$-norm does not seem to have been formally introduced? How is it defined? In the next line also $\mathcal{T}^N$ appears, which is likely a typo for $(\mathcal{T}^\pi)^N$?

**Questions:**

Refer to the box above.

---

> ### Author Response · Authors · 2024-11-26
> **Response to reviewer PF2o with comments addressed in the revised paper (part 1)**
>
> Dear reviewer, thank you for your proofreading and comments. The weaknesses you pointed out have been addressed in the revised version  and we believe that the paper is now in line with ICLR standards for acceptance and kindly ask you to change your score accordingly.
>
> > Prior Work on State Entropy.
>
> Thank you for the list of references. The introduction has been revised to provide a better overview of the literature, the limitations of existing algorithms, and better set the scope of our contribution.
>
> The novelty of our contribution has been clearly stated, and an additional experiment was added, as requested and explained here bellow.
>
> > Novelty.
>
> We have clarified the main contribution in the introduction, which is the following: we introduce a new MaxEntRL framework, i.e., a new intrinsic reward for exploration, and propose a practical algorithm for learning policies that maximize this new MaxEntRL objective.
>
> In the new introduction, we also distinguish three cases for the algorithms that maximize the entropy of the marginal discounted or marginal stationary state distributions. First, there exist model-based algorithms using a tabular representation of the transition matrix and the initial distribution. Some are off-policy and have convergence guarantees, but are limited to (small) discrete spaces. There exist algorithms that estimate the density of the state distribution with kNN, they are on-policy. Finally, other algorithms use neural density estimators for estimating the state distribution, they are arguably the best choice when large and/or continuous state and actions spaces are at hand. They are nevertheless also on-policy. We introduce a new intrinsic reward that can be estimated using a neural density estimator, which is learned off-policy using arbitrary transitions. The resulting MaxEntRL algorithm is thus also off-policy. Furthermore, the (revised) experimental results are concurrent with the existing methods.
>
> Our framework combines several advantages:
> * The MaxEntRL objective promotes (discounted) infinite-time exploration of a feature space.
> * The intrinsic reward can be estimated with a parametric function (a neural density estimator like a categorical distribution for discrete spaces, or a normalizing flow in general), and this function can be learned off-policy.
> * The learning procedure and intrinsic reward estimation is less subject to the exponential decreasing influence of future states compared to exploration with the marginal discounted state visitation measure.
>
> The claims have been adjuster to set the scope on this new reward function, and our algorithm has been benchmarked against alternative methods from the literature. The title of the paper has also been modified to better distinguish our MaxEntRL framework with existing ones.
>
> Finally, the algorithm RE3 is on-policy even if it computes the policy with SAC. At each policy update, it has to sample new trajectories to update its model of the state visitation measure. If it does not refresh the buffer, the algorithm will at best compute a policy $\pi$ that has a marginal state visitation measure $d^{\pi, \gamma}$ that maximizes the crossentropy with the distribution of states in the buffer built from a behaviour policy. It does not mean that the visitation measure is uniform. Intuitively, the policy will try to 'run away' from the states stored in the buffer.
>
> > Theoretical Ground for the Intrinsic Reward.
>
> In Theorem 1, the error on the Q-function is bounded with the conditional visitation measure instead of the error on the expected return $J$ with the marginal visitation measure, as in the papers mentioned in the review. We agree that choosing the target $q^*$ is in practice non-trivial. This theorem nevertheless allows to bound the performance of optimal exploration policies, which we believe provides intuition on why we choose this particular exploration objective. The interpretation of the theorem has been clarified in Section 3.1.
>
> Considering the link between the theorems and the practical algorithm, we have clarified the manuscript in two aspects:
> 1. In section 3.1, we clarified under what assumptions an optimal exploration policy minimizes the bound in Theorem 1. We believe it motivates our new exploration objective.
> 2. In section 3.2, we have clarified that we are going to solve a crossentropy minimization instead of minimizing a $\bar L_n$-norm. This is indeed an approximation. Nevertheless, similar surrogate optimization problems are solved in the literature when learning the conditional state visitation measures, see introduction.

---

> ### Author Response · Authors · 2024-11-26
> **Response to reviewer PF2o with comments addressed in the revised paper (part 2)**
>
> > Experiments.
>
> First, we have implemented a new concurrent algorithm maximizing the entropy of the (marginal) discounted states visitation measure. All methods are compared, and a new extensive discussion is provided in Section 4.
>
> Second, we use the min and max to highlight the best and worst cases of the different algorithms. Most importantly, it allows to clearly see that even in the worst cases, our algorithm is competitive with the best cases of the other methods. Furthermore, the median is a good performance indicator when the distribution is multimodal. Using the mean to measure the performance provides results that do not reflect any learning result.
>
> Finally, we use a discount of $0.95$ to make the long-term exploration problem even more complex. Indeed the influence of the future state probabilities is also $\gamma$-discounted in the computation of the expected return. We use truncated trajectories so that we neglect states that have negligible probabilities in the discounted visitation measure, see Appendix for hyperparameters.
>
> > Minor and Typos.
>
> We have corrected the minor errors.

---

> > ### Comment · Reviewer_PF2o · 2024-11-26
> >
> > I want to thank the authors for their thorough response and for taking reviewers' suggestions into consideration in their updated draft. From what I am reading in the response, the draft changed significantly, which will require some more time to be assessed. I will use the extended rebuttal period to do a brief pass. A few early comments below:
> > - Can the authors mark the changes in their draft in a different color? It would be great to see at first glance where it changed from the previous without having to review from the ground up;
> > - Theoretical ground for the intrinsic reward: I see that here the conditional visitation measure is concerned, but I believe the techniques used in the mentioned paper can be adapted to conditional visitations verbatim. This would reduce the significance of the presented theorem;
> > - Experimental evaluation: I am not convinced on the argument that median, max value and min value give a full picture of the experimental results. Perhaps they can be integrated (also in the appendix if space is tight) with additional plots showing other statistics, such as the mean or IQM as suggested by the paper mentioned in the review.

---

> > > ### Author Response · Authors · 2024-11-26
> > > **Response to Official Comment by Reviewer PF2o**
> > >
> > > Thank you for your response and for your help in improving the manuscript.
> > >
> > > * We have updated the manuscript to highlight the most significant changes. Other minor changes were also added to clarify some points or to extend the appendices. Thank you for considering the effort we made and taking time to proofread our manuscript.
> > >
> > > * It is likely that their results can be extended as we used similar properties to do the proof. We can add a reference to the paper for intellectual honesty. We nevertheless want to emphasize that the contribution is mostly the interpretation of the theorem, which we are still first to state as such and proof.  We believe the interpretation is impactful for our work and helps providing intuition about the fairness of our exploration objective.
> > >
> > > * We will do the computation of the different metrics. If they are equivalent, we will follow the recommendations of the paper you provided. Otherwise, we will include an appendix as you recommended.

---

> > > > ### Author Response · Authors · 2024-11-26
> > > > **Follow-up Theorem 1**
> > > >
> > > > Dear reviewer, we have updated Theorem 1 and the relative paragraph to simplify the interpretation, as requested by reviewer 6YUn. We have already added to the reference to the paper you mentioned. Please refer to the updated manuscript.

---

> ### Author Response · Authors · 2024-11-27
> **Follow-up IQM**
>
> Dear reviewer, we have computed the IQM as you suggested and results remain mostly unchanged. The only difference lies in the performance of SAC, which is slightly better in some environments. It still remains much worse than concurrent methods. Overall there is no change in interpretations and conclusion. We will follow your suggestion in the final version.

---

> > ### Comment · Reviewer_PF2o · 2024-11-30
> >
> > Dear Authors,
> >
> > Thank you for your follow-up replies. The new framing of the paper seems to better clarify the intended contribution: An off-policy algorithm for conditional entropy maximization.
> >
> > While this is interesting, I think the paper shall then compare the proposed methods in the literature, which can sometimes be used off-policy (e.g., APT by Liu & Abbeel 2021). Moreover, the additional baseline OPAC-MV seems very competitive with the CV one, if not better. The benefit of conditional visitations shall then be clarified.
> >
> > For the theoretical part, I stand with my previous opinion: The result can be implied by previous works. I think this shall be declared more clearly in the main paper.
> >
> > Best,
> >
> > Reviewer PF2o

---

> ### Author Response · Authors · 2024-12-01
> **Addressing disagreements on competitor choice, theoretical contributions, and SATO relevance**
>
> Dear Reviewer,
>
> Thank you for reviewing our paper again. We appreciated the opportunity to address your previous comments but respectfully disagree with the last message.
> 1. We have already implemented OPAC+MV, a variant of APT (Liu & Abbeel, 2021), as a competitor. In our context, we beleive that OPAC+MV is more appropriate. It uses a neural density estimator instead of kNN, which is inappropriate in discrete spaces. OPAC+MV also relies on off-policy actor-critic, which is more sample-efficient than PPO, and allows fair comparison with our algorithm. And finally, OPAC+MV works directly with the reduced and discrete state space. Since the state space is already simple, learning a new feature representation as in APT would not add value and would complicate maximizing directly the entropy of the feature space $\mathcal{Z}$ (position in the maze).
> 2. We understand the concern and can clarify in the introduction that Theorem 1 extends the results of Kakade and Langford (2002). We again emphasize the novelty of how Theorem 1 is formulated and used in the paper.
> 3. The ICLR guidelines explicitly state that achieving SOTA is not required for acceptance. Our method shows competitive results and advantages over SOTA alternatives.
>
> We believe that the paper, in its revised form, offers valuable insights and is well-motivated and validated. While we appreciate your suggestions, we feel they would not significantly enhance the quality of the paper.
> We kindly ask you to reconsider accepting this paper, as we are confident it is of interest to the ICLR community.
>
> Best regards,
>
> The authors

---

> > ### Comment · Reviewer_PF2o · 2024-12-01
> >
> > Dear Authors,
> >
> > I have realized my latest comment was too focused on the negatives and did not provide a fair account of the paper. I apologize for that and take the opportunity to clarify my evaluation: I think this project can result in a very nice contribution for the community of intrinsic motivation in RL and state entropy maximization especially. The main ideas of the paper, which from my understanding are (i) maximization of the entropy of future states visitations rather than marginal entropy, (ii) an algorithm that can work fully off-policy are both interesting and worth studying. The paper presents them with good clarity and an original eye.
> >
> > However, I believe the current manuscript does not clear the threshold for publication for two main reasons:
> > - (i) The theoretical or empirical support for the conditional entropy over marginal entropy looks somewhat weak. After going through the paper, it is not clear to me if/when I shall prefer CV over MV;
> > - (ii) While working off-policy is important to enable (more efficient) batch learning, previous approaches presented various degrees of "off-policyness" (see MEPOL by Mutti et al 2021, APT by Liu & Abbeel 2021, RE3 by Seo et al 2021). APT and RE3 sample states from the replay buffer to compute their entropy estimates, while MEPOL uses importance sampling for data reuse. How does the "off-policyness" proposed in this paper differ from theirs and why/when shall be preferred?
> >
> > I think that addressing those aspects will make for a more significant and valuable contribution.
> > Other aspects that could be addressed:
> > - Experiments could be more convincing. I agree that beating all of the baselines should not be the goal here. However, the paper only tackles a set of MiniGrid experiments (from state observation rather than images). Are those domains problematic for previous approaches? What are we learning from these results?
> > - Also related to the latter point, from the paper (and the experiments) one may think that this is a seminal work on state entropy maximization. Instead, the latter has been fairly studied in the literature. A deeper investigation on how the proposed solution relates to previous works on this topic would clarify the contribution here.
> >
> > Best regards,
> >
> > Reviewer PF2o

---

> > > ### Author Response · Authors · 2024-12-01
> > > **Clarifying choice of exploration strategy in practice and types of off-policyness**
> > >
> > > Dear Reviewer,
> > >
> > > Thank you for your response. There is no need to apologize and we value the open discussion. Below, we address your latest remarks:
> > >
> > > 1. Determining the best exploration strategy is an open question in RL. Examples can be constructed where action exploration outperforms state exploration, and inversely, in terms of the expected return of the optimal exploration policy. This question generalizes to the exploration method we propose. Without additional assumptions about the MDP, we can only bound the Q-function of an optimal exploration policy. Similarly, maximizing the entropy of marginal state distributions allows bounding the expected return using the result of Kakade and Langford (2002). However, deciding which exploration objective to favor (bounded return or Q-function, i.e. MV or CV; or even action exploration only) remains an unanswered question that is beyond this paper's scope. Nevertheless, we illustrate empirically that both entropy of marginal and conditional state distribution lead to good state-space coverage, and our method is off-policy. We will make this discussion in Section 3.1 more explicit in the manuscript.
> > > 2. The degree of “off-policyness” is a general problem in RL too. Even for policy gradient algorithms without exploration, transitioning from on-policy to off-policy using only importance sampling leads to poor algorithms (e.g., reinforce using importance sampling has high variance when the behaviour policy is too different from the optimized policy). This has motivated approaches like off-policy actor-critic, where bootstrapping in the critic learning makes the algorithm off-policy. On the one hand, our method uses bootstrapping when learning the additional function approximator to have a robust off-policy method. On the other hand, with the marginal visitation, the best we can likely do is importance sampling (which comes with the limitation discussed, and is not done in practice as it is likely very non-trivial to do when using kNN density estimation). Regarding replay buffers, they primarily reduce sample correlation and stabilize critic learning rather than inherently making an algorithm off-policy. This broader discussion lies outside our paper’s scope but we will clarify in Section 3.3 that bootstrapping is at the origin of our off-policy method.
> > > 3. The Minigrid environments we use are well-known for requiring state exploration. We believe our results, combined with the additional experiments now included in the revised manuscript, effectively illustrate the method and support our claims.
> > >
> > > As you mentioned, this discussion is highly relevant to the intrinsic reward community. With the additional emphasis on points (1) and (2), we believe our paper meets the ICLR standards. We hope this clarifies your remaining questions and kindly ask you to accept the paper.
> > >
> > > Best regards,
> > >
> > > The Authors

---

> > > > ### Author Response · Authors · 2024-12-02
> > > >
> > > > Dear Reviewer,
> > > >
> > > > Thank you again for your feedback on our paper. As the discussion period ends, we kindly ask you to reconsider increasing your score if you believe that your major concerns have been addressed.
> > > >
> > > > Best regards,
> > > >
> > > > The Authors

---

### Official Review · Reviewer_UQX8 · 2024-11-04

**Soundness:** 2
**Presentation:** 2
**Contribution:** 2
**Rating:** 3
**Confidence:** 2

**Summary:**

This paper proposes an exploration method using maximum-entropy RL in the off-policy setting. It describes the MaxEntRL framework in terms of an arbitrary feature-space, unifying state-entropy and action-entropy methods. It describes a way to compute entropy using an N-step bootstrapping operator. The main claim of the paper seems to be the novelty of unifying state-entropy and action-entropy, though I am not positive my interpretation is correct.

**Strengths:**

Maximum entropy RL is an important subject, and off-policy extensions can drastically improve sample efficiency. Using varying samples from N-step returns is an interesting and sensible idea. The two proofs in the appendix seem rigorous and correct. It’s possible that focusing on state-and-action entropy is an important direction for improving exploration, though the experimental results don't rigorously investigate this.

**Weaknesses:**

To me, the thesis of this submission is somewhat confused, and the empirical evidence undirected towards proving it. What is the core contribution exactly? Is it the N-step sampling for learning an estimator? If so, there should be experiments highlighting its effectiveness, and showing it's benefits over other approaches.  Is it the “off-policy” part? If so, that setting needs to be more clearly defined and explained. Is it the focus on S *and* A entropy rather than prior methods doing either S *o*r A? If this is the main contribution, the paper/experiments should be written to reflect that (e.g. with comparisons to only-state entropy methods). Is it a theoretical contribution of introducing the "S and A as z" framework?

Generally, I think this paper did not do an effective job of positioning itself with respect to prior work. I think the sentence “it results in the first off-policy algorithm with state and action spaces exploration” is either very incorrect or requires much more context to show it’s true. I would argue that most bonus-based exploration satisfies this requirement. For example, pseudocount-based methods at least in theory operate on (S x A), and model-prediction error definitely does. As another example, “Behavior From the Void” (Liu, Abbeel) is an off-policy state-entropy method, isn’t it?

Related, the sentence in the introduction that begins “Optimizing jointly the reward function of the MDP…” is not correct – that’s usually called bonus-based exploration. MaxEntRL can be framed as a subset of BBE, but is not its entirety.

The experimental results in this paper would be made much stronger by including relevant comparisons, such as other exploration or MaxEntRL methods. Additionally, the environments are quite simple and as such don’t convincingly demonstrate this method’s efficacy. Just comparing to vanilla SAC does not seem sufficient to me. As a small comment, the plots should have labels for the x and y axes.

Am I correct that this method is off-policy in the sense that it uses importance sampling to correct for policy-discrepancy? This is much weaker in my mind than off-policy in the Q-learning sense (only requires a dataset of transitions). Is importance sampling the central difference between this and on-policy MaxEntRL methods?

The use of $q^*$ in Equation 3 is confusing or possibly inconsistent. In Eq 3 it is just some arbitrary distribution. But then, as far as I can tell, Theorem 1 makes it seem like the distribution generated by the optimal policy. Unless it’s following established notation, using the $*$ superscript for an arbitrary (non-optimal) distribution is quite confusing.

In Equation 7, $Q^\pi(s,a)$ is undefined. Usually I wouldn’t complain because we all know what it means in standard RL, but here I’m actually unclear whether you’re talking about extrinsic discounted returns or intrinsic+extrinsic.

**Questions:**

The experiments are all effectively tabular. Is this necessary under this method to compute the objective? Can you be more clear what you mean by “the intrinsic reward equation (3) can be estimated by Monte-Carlo”? Do you need access to a generative simulator of the environment during training? If this method was applied to more complicated domains, how would it differ from for example “Behavior From the Void” (Liu, Abbeel)?

Can you clearly state the central contribution/thesis of this work, framed with respect to prior work? I had trouble understanding what the main point was, compared with parts that were in support of the main point. You cite “Marginalized State  Distribution Entropy Regularization in Policy Optimization,” which uses a similar measure of entropy – what is the essential difference between that paper and yours?

Can you clarify whether $q*$ refers to the optimal policy’s distribution in Theorem 1? If so, that should be explicit.

---

> ### Author Response · Authors · 2024-11-26
> **Response to reviewer UQX8 with comments addressed in the revised paper (part 1)**
>
> Dear reviewer, thank you for your proofreading and comments. The weaknesses you pointed out have been addressed in the revised version of the paper and the questions you asked have also been answered. We believe that the paper is now in line with ICLR standards for acceptance and kindly ask you to change your score accordingly.
>
> > What is the core contribution exactly?
>
> The main contribution is the following: we introduce a new MaxEntRL framework, i.e., a new intrinsic reward for exploration, and propose a practical algorithm for learning policies that maximize this new MaxEntRL objective.
>
> Our framework combines several advantages:
> * The MaxEntRL objective promotes (discounted) infinite-time exploration of a feature space.
> * The intrinsic reward can be estimated with a parametric function (a neural density estimator like a categorical distribution for discrete spaces, or a normalizing flow in general), and this function can be learned off-policy.
> * The learning procedure and intrinsic reward estimation is less subject to the exponential decreasing influence of future states compared to exploration with the marginal discounted state visitation measure.
>
> The contribution and limitations of the existing literature have been clarified in the revised introduction. The title of the paper has also been modified to better distinguish our MaxEntRL framework with existing ones.
>
> > Positioning with respect to prior work and off-policy algorithms.
>
> The literature review has been extended in the introduction with particular focus on the limitations of the existing algorithms. For algorithms exploring the state space, they all maximize the entropy of marginal state distributions. We distinguish three cases for the algorithms. First, there exist model-based algorithms using a tabular representation of the transition matrix and the initial distribution. Some are off-policy and have convergence guarantees, but are limited to (small) discrete spaces. There exist algorithms that estimate the density of the state distribution with kNN, e.g., "Behavior From the Void" (Liu, Abbeel), they are on-policy. Finally, other algorithms use neural density estimators for estimating the state distribution, they are arguably the best choice when large and/or continuous state and actions spaces are at hand. They are nevertheless also on-policy. We introduce a new intrinsic reward that can be estimated using a neural density estimator, which is learned off-policy using arbitrary transitions. The resulting MaxEntRL algorithm is thus also off-policy. Furthermore, the experimental results are concurrent with the existing methods.
>
> Count-based methods are a particular case of intrinsic motivation. The count (or equivalently probability of visited states during the learning procedure) is estimated on-line through learning and the algorithm is on-policy.
>
> > Bonus-based exploration and MaxEntRL.
>
> Exploration can indeed be achieved in many ways. We focus on the particular cases where an intrinsic exploration reward bonus is provided, and that exploration bonus is the entropy of some particular distribution. While less general, this definition is adopted by most of the papers we included in the introduction. We believe it is a reasonable restriction of the literature.
>
> > Experiments.
>
> Additional experiments have been added to compare our method against a MaxEntRL algorithm based on the entropy of the (marginal) discounted state visitation measure.
>
> Considering the complexity of the environments, the MiniGrid environments are well-established benchmarks. The most complex of the environment we use has $4 \times 15 \times 15 \times 15 \times 2=27000$ different possible discrete states.
>
> > Where does the off-policyness comes from? Is importance sampling the central difference between this and on-policy MaxEntRL methods?
>
> The bootstrapping operation in equation (14), where a new action is sampled from the current policy $\pi$ with probability $\gamma^N$, makes the algorithm off-policy. In practice, we neglect the importance weights, which leads to biased gradients, but keeps the method off-policy. The intuition is the same as generalized SARSA or N-step TD-learning. Note that our loss function is estimated based on transitions, not full trajectories.
>
> These elements have been clarified in the revised version of the paper. In section 3.2, the importance weights appear in the objective function. In section 3.3, we removed the importance weights in the loss function used in practice, as they are neglected, and explicitly states why the algorithm is off-policy. We also discuss the influence of neglecting the importance weights.
>
> > Clarify $q^*$ and $Q^\pi$ in Theorem 1.
>
> In theorem 1, the target $q^*$ is an arbitrary distribution, and the function $Q^\pi$ is that of the MDP without the intrinsic rewards. The discussion about Theorem 1 has been clarified in the revised version.

---

> ### Author Response · Authors · 2024-11-26
> **Response to reviewer UQX8 with comments addressed in the revised paper (part 2)**
>
> > The experiments are all effectively tabular. Is this necessary under this method to compute the objective? Can you be more clear what you mean by “the intrinsic reward equation (3) can be estimated by Monte-Carlo”? Do you need access to a generative simulator of the environment during training? If this method was applied to more complicated domains, how would it differ from for example “Behavior From the Void” (Liu, Abbeel)?
>
> The state-action space is discrete, but the function approximators are not tabular. The neural density approximator $d_\psi$ is a conditional categorial distribution. The function parameterization has been clarified in Appendix.
>
> The intrinsic reward function must be estimated and added to the extrinsic reward of the MDP to apply the RL algorithm. The intrinsic reward is a KL-divergence, which can be estimated by sampling when a generative model of the distribution $d^{\pi, \gamma}(s_\infty|s, a)$ is available. We therefore learn a model of that distribution. The approximation of the intrinsic reward is discussed in Section 3.3 and the estimate is provided in equation (15).
>
> In "Behavior From the Void"  (Liu, Abbeel) a kNN is used to estimate the entropy (and probability) of the discounted state visitation measure (not the conditional one, as we do). That method cannot be applied as such in discrete spaces. It is also computationally less efficient to do intrinsic reward predictions (due to the kNN) and it is an on-policy algorithm. As requested by the reviewer, we nevertheless implemented in the revised version a similar method to benchmark our algorithm. Details are provided in Section 4.1.
>
> > Can you clearly state the central contribution/thesis of this work, framed with respect to prior work? I had trouble understanding what the main point was, compared with parts that were in support of the main point. You cite “Marginalized State Distribution Entropy Regularization in Policy Optimization,” which uses a similar measure of entropy – what is the essential difference between that paper and yours?
>
> The main contribution is the following: we introduce a new MaxEntRL framework, i.e., a new intrinsic reward for exploration, and propose a practical algorithm for learning policies that maximize this new MaxEntRL objective.
>
> In our paper, we used the entropy of the conditional discounted state visitation measure, see equation (4) and equation (6). In the paper "Marginalized State Distribution Entropy Regularization in Policy Optimization", they use the same intrinsic reward as in "Behavior From the Void" (Liu, Abbeel), namely the marginal discounted state visitation measure. Both methods are on-policy, ours is off-policy. We implemented their algorithm as comparison to ours, and discussed the results in Section 4.
>
> The title of the paper has been modified to better distinguish our MaxEntRL framework with existing ones.
>
> > Can you clarify whether $q^*$ refers to the optimal policy’s distribution in Theorem 1? If so, that should be explicit.
>
> The target $q^*$ may be arbitrary, the discussion about Theorem 1 has been revised.

---

> ### Author Response · Authors · 2024-11-26
> **Follow-up Theorem 1**
>
> Dear reviewer, we have updated Theorem 1 and the relative paragraph to simplify the interpretation, as requested by reviewer 6YUn. Please refer to the updated manuscript. We believe that the paper is in-line with the ICLR standard for acceptance, and we once again ask you to revise your score for acceptance.

---

> > ### Author Response · Authors · 2024-12-02
> >
> > Dear Reviewer,
> >
> > Thank you again for your feedback on our paper. As the discussion period ends, we kindly ask you to reconsider increasing your score if you believe that your concerns have been addressed.
> >
> > Best regards,
> >
> > The Authors

---

### Meta-Review · Area_Chair_y4Zd · 2024-12-16

**Metareview:**

The authors propose a novel formulation of entropy maximization to encourage exploration in MDPs. However, reviewers felt that (a) the theoretical contributions were incremental or derivative, (b) experiments were not sufficiently extensive that the practical merits could be reliably communicated, and (c) the authors failed to adequately situate their contribution in the richer body of work related to maximum entropy exploration. All in all, reviewers unanimously advocated for rejection.

**Additional Comments On Reviewer Discussion:**

Reviewers were in unanimous agreement about all weaknesses of the paper.

---

### Decision · Program_Chairs · 2025-01-22

Reject